# scRNA-sequencing reveals subtype-specific transcriptomic perturbations in DRG neurons of *Pirt*EGFPf mice in neuropathic pain condition

Chi Zhang[1†], Ming-Wen Hu[2†], Xue-Wei Wang[3], Xiang Cui[1], Jing Liu[1], Qian Huang[1], Xu Cao[3], Feng-Quan Zhou[3,4], Jiang Qian[2], Shao-Qiu He[1]*, Yun Guan[1,5]*

[1]Department of Anesthesiology and Critical Care Medicine, The Johns Hopkins University School of Medicine, Baltimore, United States; [2]Department of Ophthalmology, The Johns Hopkins University School of Medicine, Baltimore, United States; [3]Department of Orthopaedic Surgery, The Johns Hopkins University School of Medicine, Baltimore, United States; [4]The Solomon H. Snyder Department of Neuroscience, The Johns Hopkins University School of Medicine, Baltimore, United States; [5]Department of Neurological Surgery, The Johns Hopkins University School of Medicine, Baltimore, United States

**Abstract** Functionally distinct subtypes/clusters of dorsal root ganglion (DRG) neurons may play different roles in nerve regeneration and pain. However, details about their transcriptomic changes under neuropathic pain conditions remain unclear. Chronic constriction injury (CCI) of the sciatic nerve represents a well-established model of neuropathic pain, and we conducted single-cell RNA-sequencing (scRNA-seq) to characterize subtype-specific perturbations of transcriptomes in lumbar DRG neurons on day 7 post-CCI. By using *Pirt*EGFPf mice that selectively express an enhanced *green fluorescent protein* in DRG neurons, we established a highly efficient purification process to enrich neurons for scRNA-seq. We observed the emergence of four prominent CCI-induced clusters and a loss of marker genes in injured neurons. Importantly, a portion of injured neurons from several clusters were spared from injury-induced identity loss, suggesting subtype-specific transcriptomic changes in injured neurons. Moreover, uninjured neurons, which are necessary for mediating the evoked pain, also demonstrated cell-type-specific transcriptomic perturbations in these clusters, but not in others. Notably, male and female mice showed differential transcriptomic changes in multiple neuronal clusters after CCI, suggesting transcriptomic sexual dimorphism in DRG neurons after nerve injury. Using *Fgf3* as a proof-of-principle, RNAscope study provided further evidence of increased *Fgf3* in injured neurons after CCI, supporting scRNA-seq analysis, and calcium imaging study unraveled a functional role of *Fgf3* in neuronal excitability. These findings may contribute to the identification of new target genes and the development of DRG neuron cell-type-specific therapies for optimizing neuropathic pain treatment and nerve regeneration.

*For correspondence:
she11@jhmi.edu (S-QiuH);
yguan1@jhmi.edu (YG)

†These authors contributed equally to this work

Competing interest: The authors declare that no competing interests exist.

## Editor's evaluation

This study identifies injury-induced changes in transcriptomic signatures in peripheral sensory neurons at a single cell level, revealing key insights into sexual dimorphism as well as plasticity-related differences between injured and uninjured neurons. These results promote an understanding of the nature of molecular events involved in the establishment of neuropathic pain.

## Introduction

The dorsal root ganglion (DRG) contains the somas of primary sensory neurons, which differ in size and axon myelination. Different gene expression profiles confer divergent neurochemical, physiologic, and functional properties on the various subtypes of DRG neurons (*Gatto et al., 2019*; *Sharma et al., 2020*; *Usoskin et al., 2015*; *Zeisel et al., 2018*; *Zheng et al., 2019*). Small-diameter neurons are important for transmitting nociceptive and thermal information, whereas large-diameter neurons are mainly non-nociceptive neurons, including mechanoreceptors and proprioceptors. Nerve injury induces various responses in DRG neurons, including cell stress, regeneration, hyperexcitability, and functional maladaptation. How these changes vary in functionally distinct neuronal subtypes and possibly affect nerve regeneration and neuropathic pain remains unclear.

Recently, single-cell/single-nucleus RNA-sequencing (scRNA-seq/snRNA-seq) has begun to reveal transcriptomic perturbations in DRG neurons after transection or crush nerve injury (*Hu et al., 2016*; *Renthal et al., 2020*). Nevertheless, there remain many important questions which are not fully addressed, especially differential transcriptional changes in functionally distinct DRG neuronal subtypes related to neuropathic pain. The crush injury model used in some of previous studies is more suitable for studying nerve regeneration than for closely capturing the etiology of clinical neuropathic pain, which often involves chronic compression, neuroinflammation, and partial injury to a major nerve. Axotomized neurons may exhibit the most profound gene expression changes that are important for regeneration. Nevertheless, neighboring uninjured DRG neurons also show significant functional changes (e.g., hyperexcitability) and contribute to dysesthesia and evoked pain hypersensitivity as a result of the remaining peripheral innervations (*Djouhri et al., 2012*; *Kalpachidou et al., 2022*; *Obata et al., 2003*; *Tran and Crawford, 2020*). Thus, identifying and differentiating transcriptional changes in injured and uninjured neurons in a cell-type-specific manner will be important to search for new targets for nerve regeneration and pain treatment. So far, most previous scRNA-seq studies have mainly focused on changes in injured neurons, but details of possible cell-type-specific transcriptomic changes in uninjured DRG neurons under neuropathic pain conditions remain partially known. Moreover, increasing clinical and preclinical evidence suggests that males and females have differences in pain sensitivity and susceptibility to chronic pain (*Fillingim et al., 2009*). To optimize clinical treatment, it will also be important to delineate sex-related gene expression changes in functionally distinct subtypes of DRG neurons after nerve injury and determine how these changes underpin sexual dimorphisms in neuropathic pain.

We established a highly efficient purification approach by using *Pirt^EGFPf* mice to enrich DRG neurons for scRNA-seq. *Pirt* is expressed in >83.9% of neurons in mouse DRG, but not in other cell types (*Kim et al., 2008*). Thus, green fluorescent protein (GFP) is selectively expressed in most DRG neurons in *Pirt^EGFPf* mice, driven by the *Pirt* promoter. GFP expression allows effective purification of DRG neurons from these mice. We then characterized perturbations of transcriptomes in DRG neurons at the single-cell level after chronic constriction injury (CCI) of the sciatic nerve. The CCI model represents a well-established neuropathic pain model that encompasses compression, ischemia, inflammation, and axonal demyelination (*Bennett and Xie, 1988*). It has been suggested to mimic the etiology of clinical conditions and induces symptoms similar to those of post-traumatic neuropathic pain in humans (*Challa, 2015*; *Costigan et al., 2010*; *Griffin et al., 2007*; *LaCroix-Fralish et al., 2011*). Since cell interactions may occur within the same ganglion, the excitability and transcriptional changes in uninjured neurons can be greatly affected by satellite glial cells (SGCs) and neighboring injured neurons. Because only a portion of sciatic nerve fibers are injured after CCI, each of the lumbar DRGs contains a mixture of injured and uninjured neurons. Thus, in addition to characterizing injured neurons which express high levels of the injury marker gene *Sprr1a*, more importantly, we also examined transcriptomic changes in uninjured neurons (*Sprr1a-*) and investigated transcriptomic sexual dimorphism after CCI at the single-cell level. Our findings may be useful for developing DRG neuron subtype-specific treatment and sex-specific therapies for nerve regeneration and neuropathic pain.

## Results

### Enrichment of DRG neurons from *Pirt^EGFPf* mice for scRNA-seq

Mice were randomly assigned to four groups (n=5 for each group): Male-CCI, Female-CCI, Male-Sham, and Female-Sham. Bilateral L4-5 DRGs were collected from mice on day 7 after bilateral sciatic

CCI or sham surgery for scRNA-seq (*Figure 1A*, *Figure 1—figure supplement 1A*). In an animal behavior study conducted on day 6 after CCI, paw withdrawal frequencies to low-force (0.07 g) and high-force (0.4 g) mechanical stimulation at the hind paws (data averaged from both sides) were significantly increased (n=5/sex), as compared to the pre-injury frequency, indicating the development of mechanical hypersensitivity (*Figure 1B*). Paw withdrawal frequencies were not significantly changed after sham surgery (n=5/sex).

DRGs contain a large number of non-neuronal cells, including SGCs, Schwann cells, immune cells, and fibroblasts. Previous studies have shown that scRNA-seq is advantageous for differentiating neurons from these non-neuronal cells (*Renthal et al., 2020*; *Wang et al., 2021*). Here, by using $Pirt^{EGFPf}$ mice, we established a highly efficient purification process to further enrich DRG neurons for scRNA-seq (*Figure 1—figure supplement 1A*).

After removing low-quality cells and doublets (*Supplementary file 2*), we recovered 3394 cells from the Male-Sham dataset; 5678 cells from the Female-Sham dataset; 2899 cells from the Male-CCI dataset; and 3681 cells from the Female-CCI dataset. *Figure 1—figure supplement 1B* showed the number of expressed genes in each dataset. We then utilized canonical correlation analysis embedded in Seurat 3.0 (1), a computational approach for minimizing experimental batch effect, to integrate cells from the four datasets for an unbiased cell clustering. The results showed that the integration worked well in our experiments, as clusters from each dataset aligned well regardless of different biological variations (*Figure 1C*). In total, we identified 16 neuronal clusters and one non-neuronal cluster (*Figure 1D*, *Figure 1—figure supplement 1C*).

The non-neuronal cluster had fewer genes and unique molecular identifier (UMI) counts than did neuronal clusters (*Figure 1—figure supplement 1D*). It expressed SGC marker genes *Fabp7* and *Apoe* (*Renthal et al., 2020*; *Wang et al., 2021*) but showed minimal expression of the pan-neuronal marker *Tubb3*. In contrast, all 16 neuronal clusters showed strong expression of *Tubb3* but had very low levels of *Fabp7* (*Figure 1—figure supplement 1E*). Although this non-neuronal cluster showed marker genes known to be expressed in both Schwann cells and SGCs (e.g., *Mpz*, *Mbp*, *Plp1*), it did not express any Schwann cell-specific genes such as *Mag*, *Prx*, or *Ncmap* (*Avraham et al., 2020*). Therefore, this non-neuronal cluster included mainly SGCs. No other non-neuronal cluster was detected in our datasets, suggesting a successful enrichment of neurons.

## Prominent new neuronal clusters appear after sciatic CCI

We further validated the identities of the neuronal clusters by a list of known subtype markers (*Figure 1E*). Our findings confirmed the presence of 12 major standard neuronal clusters (Nppb+ non-peptidergic nociceptors [NP1], Mrgprd+/Cd55+ non-peptidergic nociceptors [NP2], Mrgpra3+/Cd55+ non-peptidergic nociceptors [NP3], Tac1+/Sstr2- peptidergic nociceptors [PEP1-2], Tac1+/Sstr2+ peptidergic nociceptors [PEP3-4], Trpm8+ peptidergic nociceptors [PEP5], Trpv1+ peptidergic nociceptors [PEP6], Nefh+/Scn1b+ Aβ low-threshold mechanoreceptors [NF1, NF2], and Fam19a4+/Th+ low-threshold mechano-receptive neurons with C-fibers [cLTMR]). Importantly, we also identified four new CCI-ind1-4 clusters that were prominent in CCI groups but minimal in sham groups (*Figure 1D and E*, and *Figure 2A*). CCI-ind1-4 clusters expressed high levels of genes like *Cryba*, *Tgfbi*, *Fgf3*, *Tnfrsf12a*, *Ecel1*, and *Sema6a*, and were segregated from all standard clusters. Expression profiles of the top 50 marker genes in the 16 neuronal clusters are shown in *Figure 1F* and listed in *Supplementary file 1*.

## Gene programs in CCI-ind1-4 clusters are important to both nerve regeneration and pain

A much higher percentage of the cell population was contained in the CCI-ind1-4 clusters from CCI groups than from sham groups (*Figure 2A and B*). These clusters also showed high expression levels of injury-induced genes such as *Sprr1a and Atf3* (*Figure 2C*; *Bonilla et al., 2002*; *Tsujino et al., 2000*). In line with previous findings (*Nguyen et al., 2017*), *Atf3* was also frequently detected in the sequencing of isolated cells from sham or uninjured groups. Comparatively, *Sprr1a* was more selectively expressed in injured neurons of CCI mice (*Figure 2C*). Thus, *Sprr1a* may represent a more specific marker of injured neurons than *Aft3* in scRNA-seq (*Bonilla et al., 2002*).

Regeneration-associated genes, such as *Anxa1*, *Flnc*, *Gadd45a*, *Inhbb*, *Itga7*, *Kif22*, *Plaur*, *Sema6a*, *Sox11*, *Tnfrsf12a*, and *Tubb6*, were among the top 30 marker genes of CCI-ind1-4 clusters (*Figure 2D*;

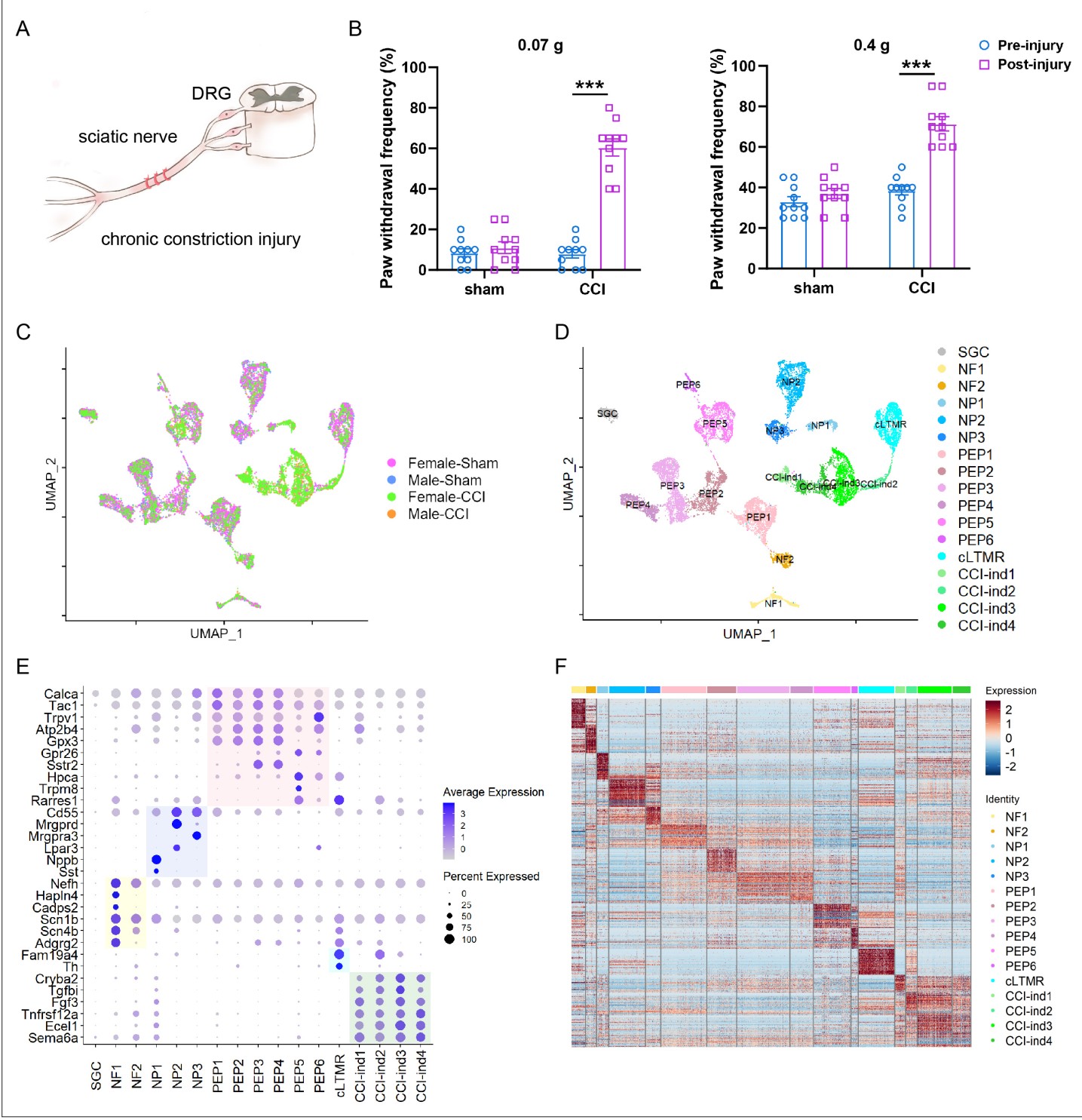

**Figure 1.** Single-cell RNA-sequencing (scRNA-seq) identified distinct clusters of cells in the dorsal root ganglion (DRG) of *Pirt^EGFPf* mice. (**A**) Schematic diagram showing the procedure for chronic constriction injury (CCI) of the sciatic nerve. (**B**) Paw withdrawal frequencies to low-force (0.07 g von Frey filament, left) and high-force (0.4 g, right) mechanical stimuli before and 6 days after CCI or sham surgery. n=10 per group (n=5/sex). Two-way mixed-model analysis of variance (ANOVA) followed by Bonferroni post hoc test. Data are expressed as mean ± SD, ***p<0.001 versus pre-injury. (**C**) Integration of four datasets visualized by uniform manifold approximation and projection (UMAP). (**D**) Seventeen distinct cell clusters were identified by Seurat, including SGC (1), NF (2), NP (3), PEP (6), cLTMR (1), and CCI-induced clusters (4). (**E**) Dot plot of subtype-specific marker genes in each cluster. Genes highlighted in the yellow, purple, pink, and blue zones are known markers for NF, NP, PEP, and cLTMR, respectively. Genes highlighted in the green zone are markers identified in CCI-induced (CCI-ind) clusters. The dot size represents the percentage of cells expressing the gene, and the

*Figure 1 continued on next page*

*Figure 1 continued*

color scale indicates the average normalized expression level in each cluster. (**F**) A heatmap shows the expression patterns of the top 50 marker genes in each cluster.

The online version of this article includes the following figure supplement(s) for figure 1:

**Figure supplement 1.** Data quality assays.

*Chandran et al., 2016*). Intriguingly, some of them may also be involved in neuropathic and inflammatory pain. For example, animal studies suggest that neurotensin (encoded by *Nts*) (*Guillemette et al., 2012*; *Sarret et al., 2005*) and annexin1 (encoded by *Anxa1*) may attenuate neuropathic pain and inflammatory pain, respectively (*Pei et al., 2011*; *Zhang et al., 2021*). *Nnat* and *Sdc1* were exclusively increased in nociceptive DRG neurons after nerve injury (*Chen et al., 2010*; *Murakami et al., 2015*). *Sox11* was identified by integrated bioinformatic analysis as a novel gene that is essential to neuropathic pain (*Chen et al., 2021*).

Gene ontology (GO) analysis of the top 50 marker genes showed that CCI-induced transcriptomic changes were important for both nerve regeneration (e.g., nervous system development, axon guidance, cellular response to nerve growth factor stimulus) and neuronal excitability (e.g., positive regulation of calcium ion import, response to pain, regulation of sodium ion transport, and the neuropeptide signaling pathway; *Figure 2E*).

## NP1, PEP5, NF1, and NF2 clusters exhibit different transcriptional programs from other clusters after CCI

Previous studies in different nerve injury models showed that no specific neuronal cluster was spared from injury. Injured neurons in all clusters lost their original subtype-specific marker genes beginning at day 1 after injury, and hence could no longer be categorized into original clusters (*Hu et al., 2016*; *Nguyen et al., 2019*; *Renthal et al., 2020*; *Wang et al., 2021*). Yet, these studies did not further examine the proportion of injured neurons in each cluster after injury. Our findings showed a large decrease of cell proportion in 8 of 12 standard neuronal clusters after CCI (*Figure 3—figure supplement 1*), suggesting that the injured neurons in these clusters were no longer categorized into their original clusters. Instead, they may be assigned to CCI-ind1-4 clusters (*Figure 3A and B*), as suggested by a previous study (*Renthal et al., 2020*). The remaining four clusters (NP1, PEP5, NF1, NF2) showed little decrease in cell proportion after CCI. Moreover, a portion of injured neurons (*Sprr1a*+) in these four clusters were still categorized into their original clusters after CCI (*Figure 3A and B*). Further analysis revealed two subpopulations of neurons in these clusters (*Figure 3C–F*). For example, in the NP1 cluster, the larger subpopulation showed high expression of *Sprr1a* and other injury-induced genes (*Gal*, *Hspb1*, *Stmn4*, *Chl1*, and *Sox11*) (*Chandran et al., 2016*), suggesting that these are injured neurons (*Figure 3C*). In contrast, the smaller subpopulation (*Sprr1a*-) showed little or no expression of injury-induced genes, but expressed other genes highly, such as *Nppb* and *Sst*, which are NP1 subtype-specific marker genes (*Figure 1E*). We found similar results in PEP5, NF1, and NF2 clusters (*Figure 3D–F*).

## Transcriptional changes in injured neurons of NP1, PEP5, NF1, and NF2 clusters after CCI

Because a significant portion of injured neurons (*Sprr1a*+) in NP1, PEP5, NF1, and NF2 clusters maintained their identities after CCI (*Figure 3*), we were able to determine transcriptomic changes in these neurons by comparing them to uninjured neurons of the same clusters in the sham group. After CCI, 197, 67, 41, and 79 differentially expressed genes (DEGs) were generated from NP1, PEP5, NF1, and NF2 clusters, respectively (*Supplementary file 3*), with 19 shared DEGs (*Supplementary file 4*). Most were common regeneration-associated genes induced by nerve injury (*Chandran et al., 2016*), and some (*Cacna2d1*, *Gal*, *Gap43*, *Gadd45a*, *Atf3*, *Sprr1a*) were also significantly regulated under chronic pain conditions (*LaCroix-Fralish et al., 2011*; *Perkins et al., 2014*).

GO analysis showed that these four clusters shared many common pathways, including those related to nervous system development, axon guidance, neuron projection development, microtubule-based process, neuropeptide signaling pathway, and cell differentiation (*Figure 4A*). In addition, we observed CCI-induced changes that affect neuronal excitability (e.g., downregulation of potassium

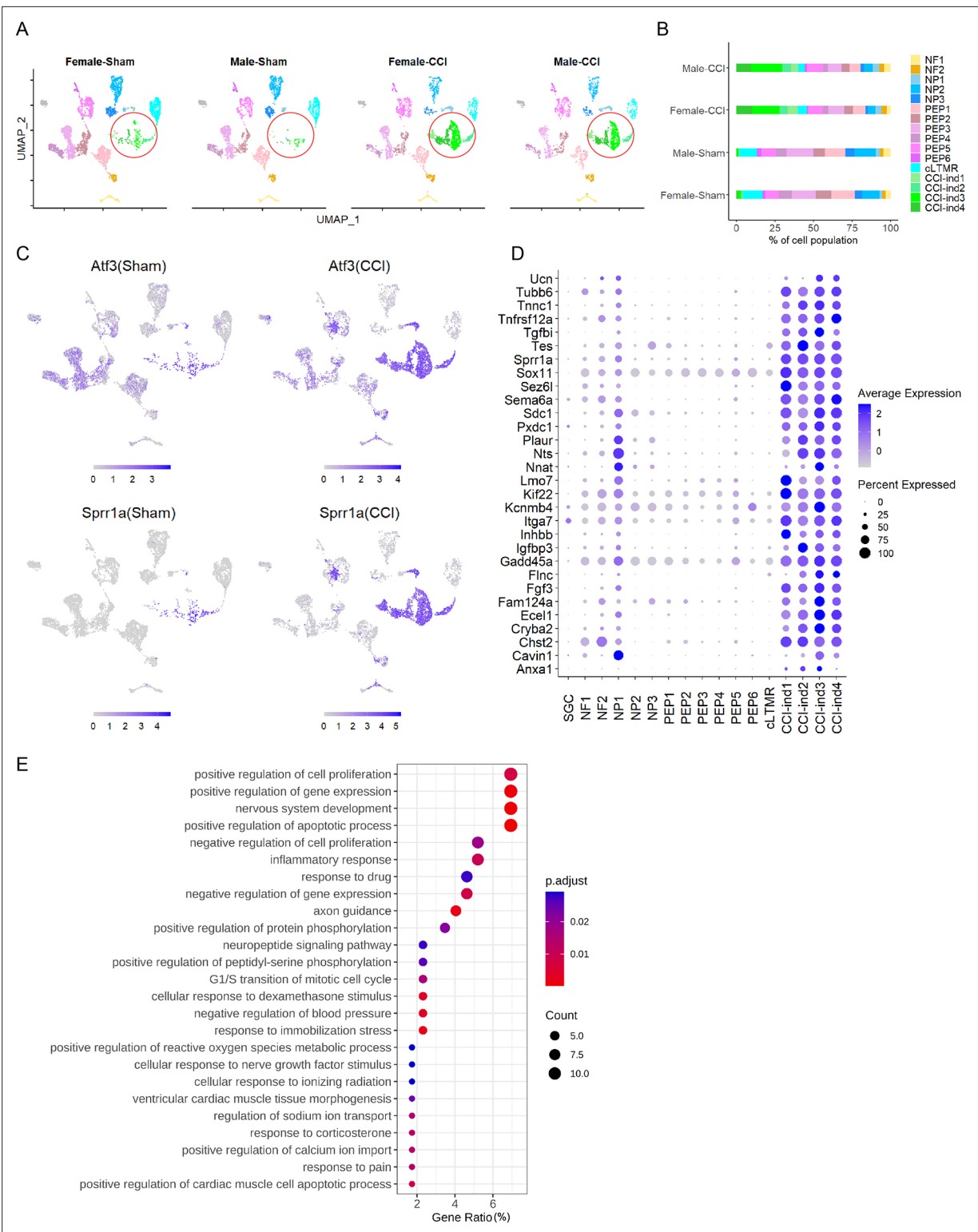

**Figure 2.** New neuronal clusters are induced by chronic constriction injury (CCI) of the sciatic nerve. (**A**) Sciatic CCI induced four new clusters (marked in a red circle) of dorsal root ganglion (DRG) neurons in both female and male mice. These new clusters were named CCI-induced (CCI-ind) 1, 2, 3, and 4, and were not prominent in sham groups. x-axis: uniform manifold approximation and projection 1 (UMAP1), y-axis: UMAP2. (**B**) Percentage of cell population in 16 neuronal clusters present in each of the four treatment groups. (**C**) Feature heatmap shows the expression levels of injury-induced genes (*Atf3, Sprr1a*) in different clusters of CCI and sham groups. (**D**) The dot plot shows the top 30 marker genes of CCI-ind1-4 clusters, as compared to those in other clusters. (**E**) Top 25 biological processes enriched by the top 50 marker genes from CCI-ind1-4 clusters.

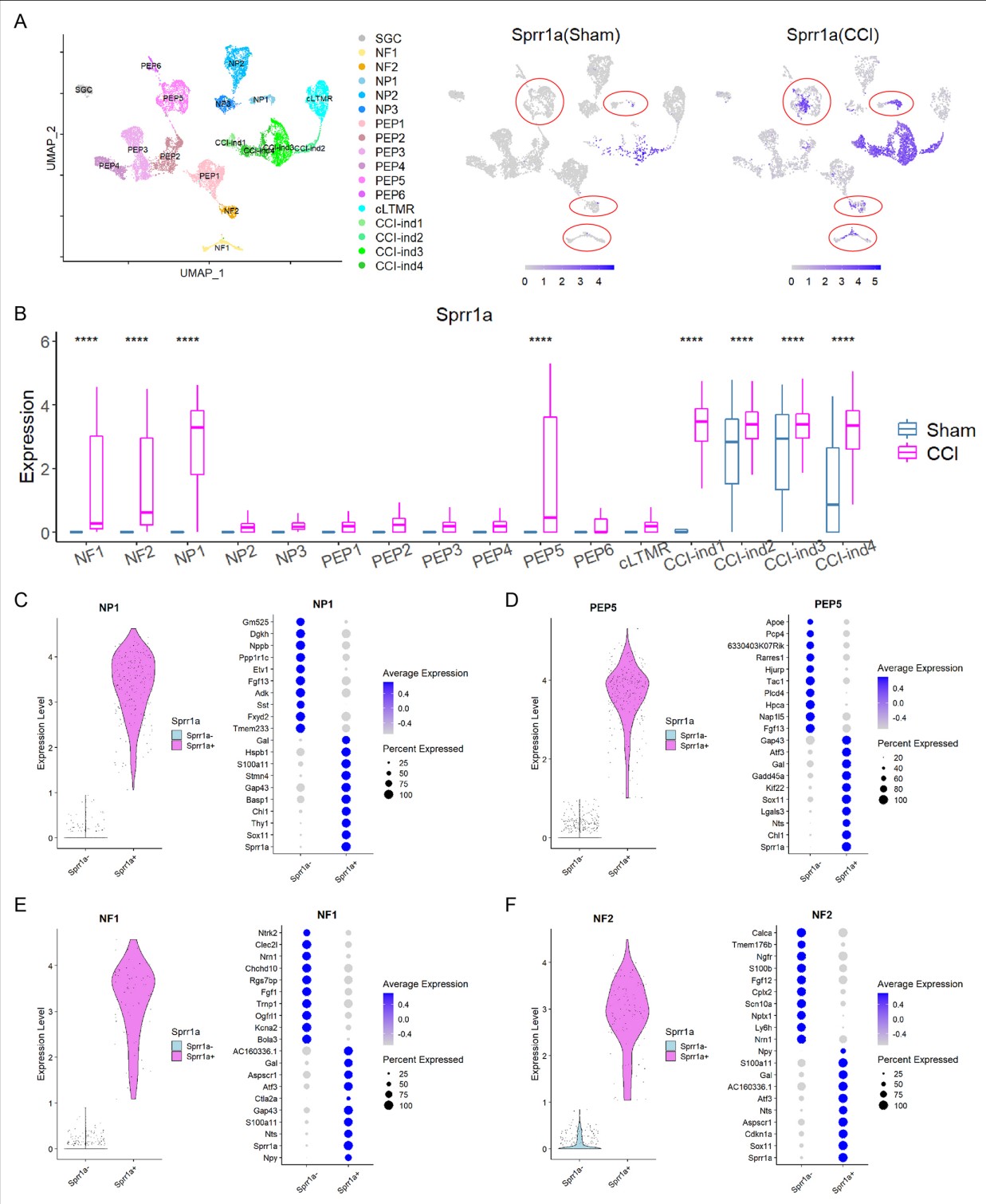

**Figure 3.** Transcriptional program changes in different neuronal clusters after sciatic nerve chronic constriction injury (CCI). (**A**) Left: The identities of 17 clusters of dorsal root ganglion (DRG) cells visualized by uniform manifold approximation and projection (UMAP). Right: UMAP displays distinct expression patterns of *Sprr1a*, an injury-induced gene, in cells of NP1, PEP5, NF1, and NF2 clusters (indicated by red circles). Each of these clusters contained *Sprr1a*⁺and *Sprr1a*⁻ cells, resulting in two subpopulations. (**B**) Box plots show that *Sprr1a* expression was statistically different between sham and CCI groups in only four standard neuronal clusters (NP1, PEP5, NF1, and NF2), suggesting a subtype-specific expression profile in sham and CCI groups. We selected clusters (NF1, NF2, NP1, PEP5, CCI-ind1, CCI-ind2, CCI-ind3, CCI-ind4) with normalized unique molecular identifier (UMI) expression >1 in sham and CCI conditions for statistical analysis. n=330 (NF1), 180 (NF2), 190 (NP1), 806 (PEP5), 28 (CCI-ind1), 70 (CCI-ind2), 138 (CCI-

*Figure 3 continued on next page*

*Figure 3 continued*

ind3), and 64 (CCI-ind4) for clusters in sham condition, and n=202, 215, 258, 631, 369, 351, 1188, 638 for these clusters in CCI condition. Student's t-test, ****p<0.0001 versus sham. (**C–F**) Left: Two subpopulations (*Sprr1a+*, *Sprr1a-*) of cells in NP1 (**C**), PEP5 (**D**), NF1 (**E**), and NF2 (**F**) clusters were separated based on the expression of *Sprr1a*. Right: Top 10 DEGs in *Sprr1a+* and *Sprr1a-* subpopulations of NP1, PEP5, NF1, and NF2 clusters.

The online version of this article includes the following figure supplement(s) for figure 3:

**Figure supplement 1.** Cell number of each neuronal cluster in the four datasets.

and sodium channels, upregulation of calcium channel Cacna2d1, dysregulation of genes encoding neuropeptide, and G-protein-coupled receptors; *Figure 4—figure supplement 1*).

We further examined pain-related protein-protein interaction (PPI) networks within the pain inter-actome, a comprehensive network of 611 interconnected proteins specifically associated with pain (*Jamieson et al., 2014*). Examining 197 DEGs of the NP1 cluster revealed an interconnected network of 93 genes (*Figure 4B*). Among them, *Calca, Prkca, Sst*, and *Tac1* were hub genes that were significantly downregulated after CCI, whereas *Tgfb1* and *Gal* were hub genes that were significantly upregulated. Intriguingly, the top marker gene of the NP1 cluster, *Sst*, is also a key gene for neuropathic pain (*Zhu et al., 2019*); hence it may be an important new target for pain modulation.

When we examined an interconnected network of 44 genes from 67 DEGs in the PEP5 cluster, we identified *Jun, Gal, Nts*, and *Tac1* as hub genes that changed significantly after CCI (*Figure 4C*). Examination of 41 DEGs from the NF1 cluster revealed an interconnected network of 69 genes (*Figure 4D*), with *Calca, Npy, Jun, Gal*, and *Nts* as hub genes. Similarly, an examination of 79 DEGs from the NF2 cluster revealed an interconnected network of 53 genes (*Figure 4E*), including hub genes *Npy, Jun, Gal, Nts*, and *Tac1*.

Among these hub genes, *Calca* and *Tac1* were downregulated whereas *Npy, Gal, Jun*, and *Nts* were upregulated in multiple clusters. Functionally, *Npy* was recently identified as a key prognostic and therapeutic target of neuropathic pain (*Tang et al., 2020*), and *Jun* and *Nts* may play pivotal modulatory roles in both neuropathic pain and nerve regeneration (*Zhao et al., 2020*). For example, intrathecal administration of neurotensin (encoded by *Nts*) induced pain inhibition in animal models of neuropathic pain (*Guillemette et al., 2012*; *Sarret et al., 2005*). It is possible that the upregulation of *Nts* may represent a compensatory change after injury to limit the exaggeration of neuropathic pain. Future studies are warranted to delineate the roles of each of these shared hub genes in neuropathic pain pathogenesis.

## A subset of neuronal clusters shows subtype-specific transcriptional changes in uninjured neurons

Another goal of our study was to explore transcriptional changes in uninjured neurons (*Sprr1a-*) of different clusters under neuropathic pain conditions. Strikingly, many DEGs were identified in *Sprr1a-* neurons from a subset of clusters including NP1 (143), PEP5 (232), NF1 (95), and NF2 (132) after CCI (*Figure 5A*, *Supplementary file 5*). Comparatively, only a few DEGs were present in *Sprr1a-* neurons from other clusters, suggesting subtype-specific transcriptional changes in uninjured neurons.

GO analysis showed that top pathways shared by *Sprr1a-* neurons in NP1, PEP5, NF1, and NF2 clusters are related to protein biosynthetic processes, including translation, proton transport, ribosomal small subunit assembly, and transport (*Figure 5B–F*, *Supplementary file 6*). Pathways related to nerve regeneration and neuropathic pain were also found in these clusters but were much less significant. These findings suggest that these uninjured neurons in NP1, PEP5, NF1, and NF2 clusters may enter a 'preparation' state in response to nerve injury.

## Sex differences in transcriptional changes of different DRG neuronal subtypes after CCI

Both human and animal models suggest the presence of sex differences in pain sensitivity and chronic pain prevalence (*Fillingim et al., 2009*; *Pieretti et al., 2016*). The peripheral neuronal mechanisms underlying these sexual dimorphisms remain unclear, and few studies have compared transcriptional changes of DRG neurons at the single-cell level, especially under neuropathic pain conditions. *X inactive-specific transcript* (*Xist*) is a specific transcript expressed exclusively by the inactive X chromosome in female mammals (*Borsani et al., 1991*). Consistent with previous findings, we did not

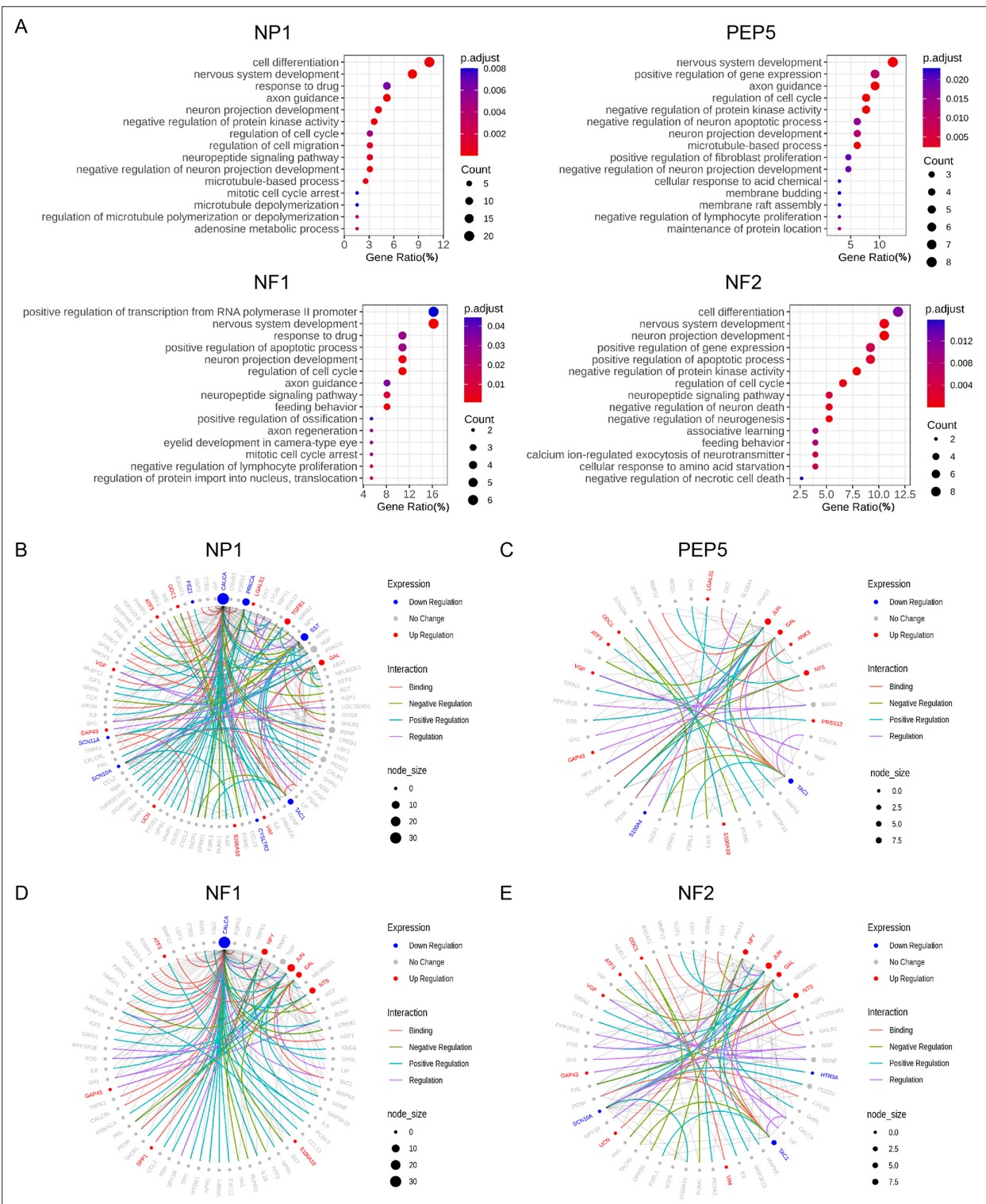

**Figure 4.** Gene ontology analysis of chronic constriction injury (CCI)-induced differentially expressed genes (DEGs) and pain-related protein-protein interaction (PPI) networks in NP1, PEP5, NF1, and NF2 clusters. (**A**) Gene ontology analysis of biological processes enriched by CCI-induced DEGs in NP1, PEP5, NF1, and NF2 clusters. (**B–E**) The neuropathic pain-specific PPI networks of CCI-induced DEGs in NP1, PEP5, NF1, and NF2 clusters. Colored edges mark the type of interaction. Colored nodes mark the expression changes after CCI. Node size indicates the number of interactions against pain interactome.

The online version of this article includes the following figure supplement(s) for figure 4:

*Figure 4 continued on next page*

*Figure 4 continued*

**Figure supplement 1.** Chronic constriction injury (CCI) altered the expression of genes that encode ion channels, neuropeptides, and G-protein-coupled receptors (GPCRs) in dorsal root ganglion (DRG) neurons.

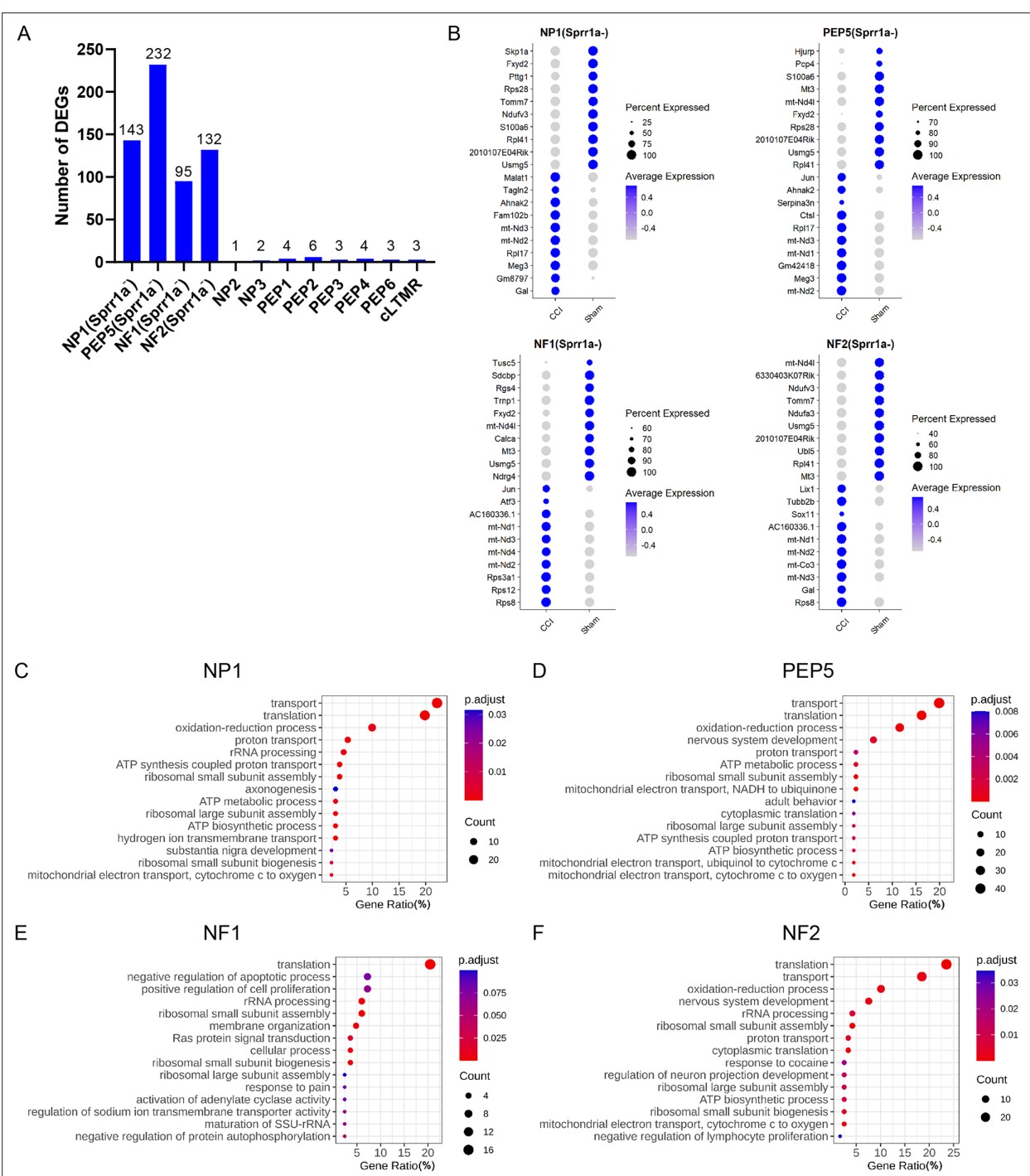

**Figure 5.** Gene ontology analysis of chronic constriction injury (CCI)-induced differentially expressed genes (DEGs) in the *Sprr1a*⁻ subpopulation of NP1, PEP5, NF1, and NF2 clusters. (**A**) The bar graph shows the number of DEGs induced by CCI in *Sprr1a*⁻ neurons of each cluster. (**B**) Top 10 marker genes of *Sprr1a*⁻ neurons in NP1, PEP5, NF1, and NF2 clusters of CCI and sham groups. (**C–F**) Gene ontology analysis of CCI-induced DEGs in *Sprr1a*⁻ neurons in NP1 (**C**), PEP5 (**D**), NF1 (**E**), and NF2 (**F**) clusters.

detect *Xist* expression in cells from our male mice (***Figure 6—figure supplement 1A***). Cell subtype distributions were similar between female and male mice in both sham and CCI groups (***Figure 2B***). Furthermore, Female-Sham and Male-Sham groups showed good correlation in cluster comparison, indicating a great similarity of transcriptional programs under physiologic conditions (***Figure 6—figure supplement 1B***). Nevertheless, our findings suggest sex differences in the transcriptional changes that occurred after CCI. When compared to the corresponding sham group, the Male-CCI group and

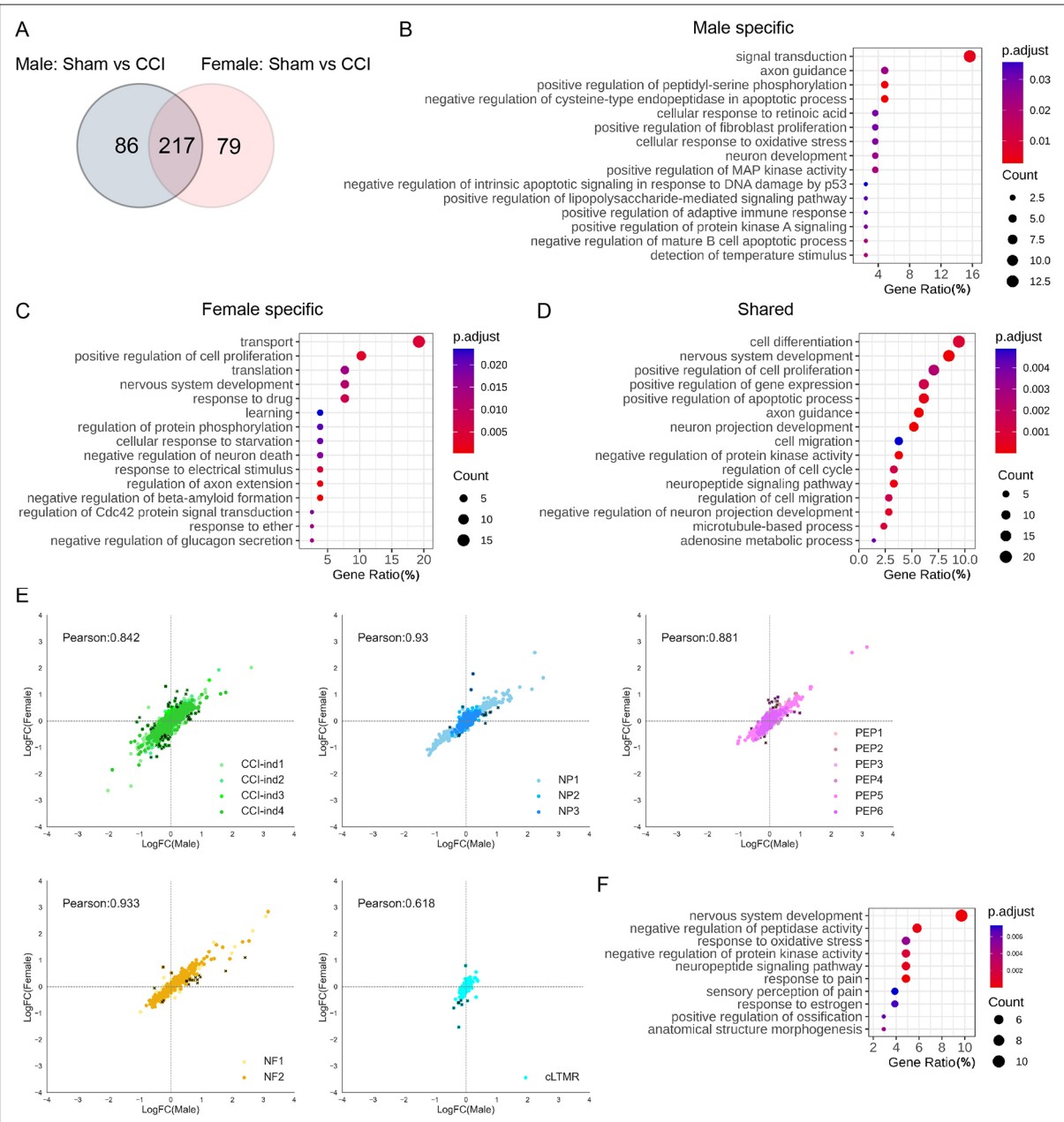

**Figure 6.** Comparisons of transcriptional changes between female and male mice after chronic constriction injury (CCI). (**A**) The Venn diagram shows the number of genes that were differentially expressed between CCI and sham in male and female mice. (**B–D**) Gene ontology pathways that are associated with differentially expressed genes (DEGs) in male mice (**B**), female mice (**C**), and both male and female mice (**D**). (**E**) Pearson correlations based on the fold-change of 382 DEGs after CCI in CCI-ind clusters, NP, PEP, NF, and cLTMR. Black dots represent 106 genes that showed >2-fold differences between female and male mice. (**F**) Gene ontology analysis of the 106 genes from panel E.

The online version of this article includes the following figure supplement(s) for figure 6:

**Figure supplement 1.** Comparisons of specific differentially expressed genes (DEGs) between female and male mice after chronic constriction injury (CCI).

the Female-CCI group exhibited 303 and 296 DEGs, within which there were 79 female-specific and 86 male-specific DEGs (*Figure 6A*, *Supplementary file 7*). Top pathways were further identified by conducting GO analysis of sex-specific and shared DEGs (*Figure 6B–D*).

Pearson correlation analysis was also performed based on the fold-change of the combined 382 DEGs. Most neuronal clusters showed a good correlation of DEGs between males and females (*Figure 6E*), indicating similarity and a minimal batch effect. Yet, the cLTMR cluster had a poor correlation. Intriguingly, Bohic et al. reported that deletion of *Bhlha9*, a transcription factor which is highly expressed in cLTMR, impaired thermotaxis behavior and exacerbated formalin-evoked pain only in male mice (*Bohic et al., 2020*). Furthermore, *Calca*, which is a nociceptor-specific gene that is highly upregulated in cLTMRs of male *Bhlha9*-null mice (*Bohic et al., 2020*), showed >2-fold differences between female and male cLTMRs in our dataset. These findings suggest that cLTMRs may play an important role in the sexual dimorphism of pain. From all clusters, 106 genes showed >2-fold differences between female and male CCI mice (*Figure 6E*, *Supplementary file 8*). The top 10 pathways from the GO analysis of these DEGs included nervous system development, neuropeptide signaling pathway, response to pain, sensory perception of pain, and response to estrogen (*Figure 6F*). Collectively, these findings suggest differential transcriptomic changes in DRG neurons after CCI between the two sexes.

## Distribution of increased *Fgf3* expression in DRG neurons after CCI

*Fgf3* is one of the top upregulated genes in CCI-ind1-4 clusters which express high levels of injury-induced gene *Sprr1a* as shown in our scRNA-seq study (*Figures 1 and 2*, *Supplementary file 1*). Previous studies suggested that *Fgf3* may play a role in pain and neuropathy. For example, mice lacking NMDA receptor GluN1 in Schwann cells exhibited pain hypersensitivity and upregulated *Fgf3* and *Sprr1a* expression in DRG neurons (*Brifault et al., 2020*). Accordingly, we conducted RNAscope in situ hybridization to further examine the expression and distribution of *Fgf3* in DRG sections and compared findings between sham-operated and CCI mice. Both *Fgf3* and *Sprr1a* signals were minimal in the sham group, but robustly increased after CCI (*Figure 7A–C*). In CCI mice, a large portion of *Sprr1a*+ cells were *Fgf3*+, suggesting the expression of *Fgf3* in injured neurons. In contrast, *Sprr1a*- neurons which express different cell subtype markers (*Tac1*, *Nefh*, *Nppb*) were rarely *Fgf3*+ in both CCI and sham groups (*Figure 7D*).

We next conducted an integration analysis of our scRNA-seq datasets and Renthal's snRNA-seq datasets. The integration of six datasets (Renthal's datasets: Naïve, Crush; our datasets: Male-Sham, Female-Sham, Male-CCI, Female-CCI) and twenty-two distinct clusters (Renthal's datasets: cLTMR, NF, NP, PEP, SST; our datasets: SGC, NF1-2, NP1-3, PEP1-6, cLTMR, and CCI-ind1-4 clusters) were visualized by uniform manifold approximation and projection (UMAP) (*Figure 7—figure supplement 1A and B*). Feature heatmaps showed that *Fgf3* expression was increased mainly in injury-induced new neuronal clusters (indicated by red circle) which were prominent in three injury datasets/groups (Crush, Male-CCI, Female-CCI), but were minimal in control groups (Naïve, Male-Sham, Female-Sham, *Figure 7—figure supplement 1C*). Collectively, evidence from in situ hybridization study and results from the integration analysis support our scRNA-seq findings, suggesting that *Fgf3* mRNA expression was increased mainly in the injured neurons which were *Sprr1a*+ and negative for subtype markers after nerve injury.

We also did a correlation analysis between our CCI-ind1-4 clusters and neuronal clusters identified in Renthal's datasets (7 days after sciatic nerve crush) and found that the CCI-ind1-4 clusters showed a good correlation with cLTMR1, cLTMR2, and NP in their datasets (*Figure 7—figure supplement 2*).

## Functional examination of *Fgf3* in DRG neurons

We then conducted in vitro calcium imaging to examine whether attenuating the upregulated *Fgf3* expression may alter DRG neuronal response to capsaicin, a TRPV1 agonist that induces heat pain, in CCI mice. Lumbar DRG neurons from sham and CCI mice were transfected with siRNA specifically targeting *Fgf3* (si*Fgf3*, 0.2 nmol) or non-targeting siRNA (siNT, control) in culture. Previously we have demonstrated that the transfection efficiency of small RNA oligos using nucleofection was over 90% (*Jiang et al., 2015*). Quantitative PCR study showed that *Fgf3* mRNA level in siNT-transfected DRG neurons was significantly higher in CCI group (n=3) than that in sham group (n=2, *Figure 7E*), which is

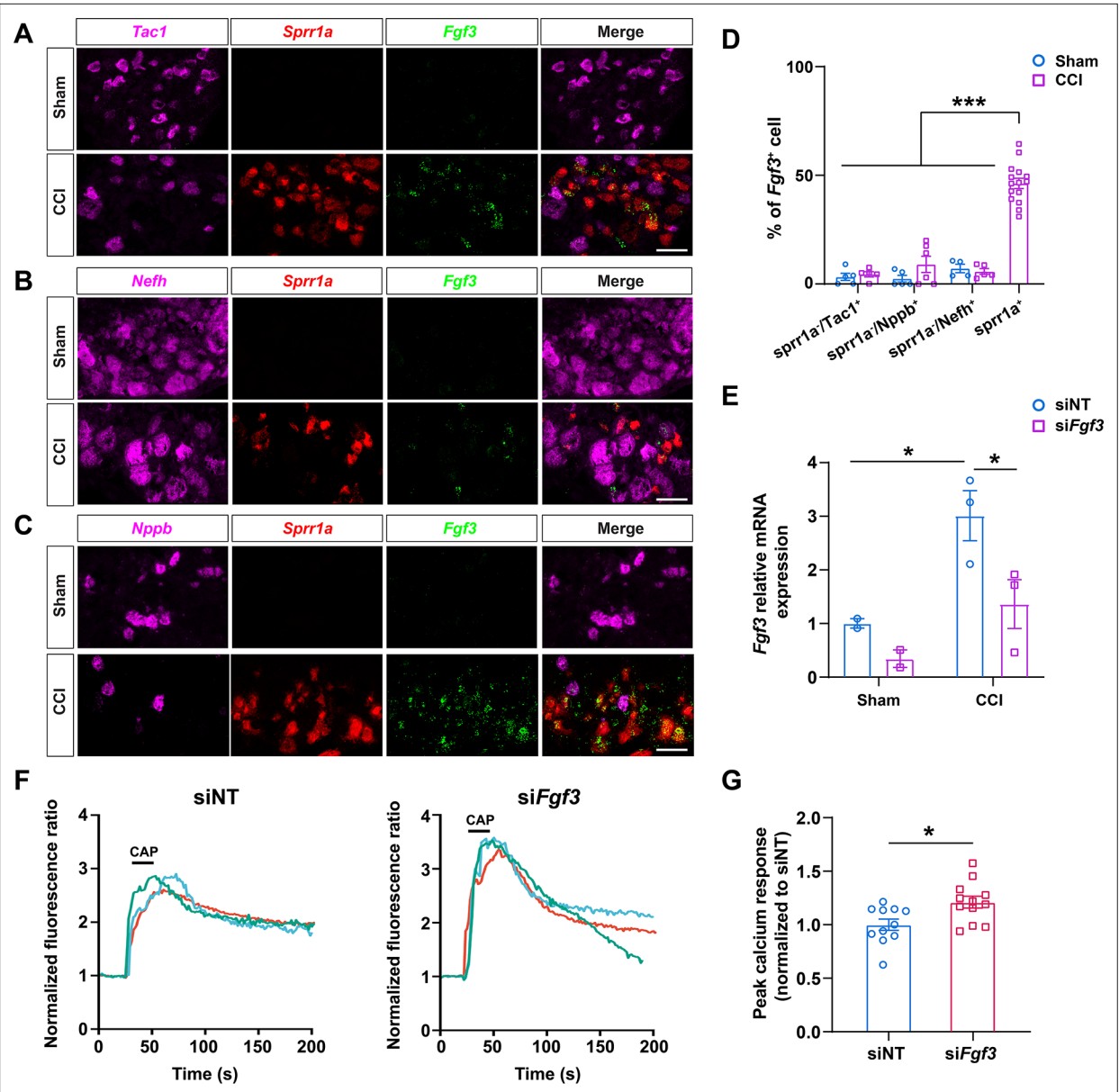

**Figure 7.** The distribution and functional examination of *Fgf3* in mouse dorsal root ganglion (DRG) neurons. (**A–C**) Representative RNAscope in situ hybridization images of lumbar DRGs from sham-operated (Sham) and chronic constriction injury (CCI) mice stained with probes against *Fgf3* (green), *Sprr1a* (red, an injury marker), and different cell subtype markers *Tac1*(**A**), *Nefh* (**B**), *and Nppb* (**C**) (magenta). Scale bar, 50 μm. (**D**) Quantification of labeled cells as the percentage of *Fgf3*+ cells. (n=316, 278, 106 for *Sprr1a*-/*Tac1*+, *Sprr1a*-/*Nefh*+, *Sprr1a*-/*Nppb*+ cells from four sham-operated mice; n=252, 360, 88 for *Sprr1a*-/*Tac1*+, *Sprr1a*-/*Nefh*+, and *Sprr1a*-/*Nppb*+ cells, and n=1771 for *Sprr1a*+ cells from four CCI mice). One-way analysis of variance (ANOVA) followed by Bonferroni post hoc test. Data are expressed as mean ± SEM, ***p<0.001 versus indicated groups. (**E**) The levels of *Fgf3* mRNA in cultured DRG neurons from sham-operated (n=2) and CCI mice (n=3) were assayed by qPCR. DRG neurons were transfected with siRNA specifically targeting *Fgf3* (si*Fgf3*, 0.2 nmol) or non-targeting siRNA (siNT) as control. Two-way ANOVA followed by Bonferroni post hoc test. Data are expressed as mean ± SEM, *p<0.05 versus siNT in sham group and si*FGF3* in CCI group. (**F**) Representative traces of calcium responses to capsaicin (0.3 μM, bath application) in cultured DRG neurons from CCI mice. Before in vitro calcium imaging, DRG neurons were treated with either si*Fgf3* (0.2 nmol) or siNT control. (**G**) The quantification of evoked calcium responses to capsaicin in each group (si*Fgf3*: n=12 coverslips with 263 neurons; siNT: n=11 coverslips with 193 neurons). Student's t-test. Data are expressed as mean ± SEM, *p<0.05 versus siNT.

The online version of this article includes the following figure supplement(s) for figure 7:

**Figure supplement 1.** The *Fgf3* expression was increased in nerve injury-induced new neuronal clusters.

**Figure supplement 2.** Correlation analysis between dorsal root ganglion (DRG) neuronal clusters identified in our study and those from Renthal's study.

consistent with our scRNA-seq and in situ hybridization results. Compared to siNT, si*Fgf3* transfection significantly decreased the upregulated *Fgf3* mRNA level in CCI group (*Figure 7E*).

Because the expression of *Fgf3* was quite low in sham group, we conducted in vitro calcium imaging on cultured DRG neurons from CCI mice. In CCI group, the intracellular calcium rise evoked by capsaicin (0.3 μM, bath application) was significantly increased in neurons transfected with si*Fgf3*, as compared to siNT (*Figure 7F*).

## Discussion

Primary sensory neurons are the fundamental units of the peripheral sensory system and are important for pharmacologic treatment of pain and sensory nerve regeneration. Here, we provided new insights into subtype-specific transcriptomic changes in DRG neurons under neuropathic pain conditions. First, in addition to 12 standard clusters validated by known neuronal subtype marker genes, four CCI-ind clusters devoid of subtype marker genes showed a strong presence after CCI. These findings support recent observations by *Renthal et al., 2020*. Second, we unraveled four neuronal clusters (NP1, PEP5, NF1, NF2) which contain both uninjured (*Sprr1a*-) and injured (*Sprr1a*+) subpopulations after CCI whereas injured neurons of other clusters were mostly segregated into CCI-ind clusters as unidentified cells after CCI. Importantly, we provided novel evidence that uninjured neurons of these clusters also exhibited subtype-specific transcriptomic perturbations after CCI. Third, our gene expression program uncovered sex differences in the transcriptional changes of DRG neurons after CCI at the single-cell level. Lastly, using *Fgf3* as a proof-of-principle, our RNAscope in situ hybridization study showed increased expression and distribution of *Fgf3* in DRG neurons after CCI, supporting scRNA-seq analysis, and in vitro calcium imaging study further suggested a functional role of *Fgf3* in neuronal excitability.

Genome-wide screening on bulk DRG tissues has demonstrated profound transcriptional changes after nerve injury (*Chandran et al., 2016*; *LaCroix-Fralish et al., 2011*). Yet, because bulk DRG tissue includes a mixture of different neuronal subtypes and non-neuronal cells, bulk RNA-seq cannot distinguish differential transcriptional changes that occur in specific cell subtypes. Recently, scRNA-seq and snRNA-seq studies have begun to uncover subtype-specific perturbations of gene expression in DRG after nerve injury (*Nguyen et al., 2019*; *Renthal et al., 2020*; *Wang et al., 2021*). However, most previous studies isolated cells or nuclei without effectively enriching neurons for sequencing. Consequently, a large number of non-neuronal cells would undergo sequencing, thereby reducing sequencing depth of neurons (*Renthal et al., 2020*; *Wang et al., 2021*). By using *Pirt*^EGFPf mice in which EGFP is selectively expressed in most DRG neurons, we improved the purification and successfully enriched DRG neurons for scRNA-seq, with only a few residual SGCs; most other non-neuronal clusters were excluded. The resulting increase in sequencing depth largely increased the number of genes detected in DRG neurons, particularly for genes with low read counts (*Perkins et al., 2014*). It needs to be noted that although *Pirt* is expressed in >83.9%of DRG neurons, the remaining *Pirt-negative* neurons are mainly NF200+ neurons and have large-diameter cell bodies (*Kim et al., 2008*). Thus, the NF population may be slightly underrepresented in our samples.

Our clustering analysis identified 16 distinct neuronal clusters and one SGC cluster. Of those, 12 standard neuronal clusters were categorized based on known subtype marker genes, which were present in both sham and CCI groups. These findings suggest that a subpopulation of neurons in each subtype was spared from injury and maintained its distinguishing transcriptional program at day 7 post-CCI (*Bennett and Xie, 1988*). Strikingly, CCI-ind1-4 clusters showed diminished expression of subtype marker genes but high expression of injury-induced genes (*Atf3*, *Sprr1a*). These clusters were present at high levels only after CCI. Accordingly, they are likely injured neurons that have lost their original subtype marker genes. Indeed, the top 50 DEGs in CCI-ind1-4 clusters included those important to nerve regeneration and neuronal hyperexcitability. Because these neurons are likely axotomized and disrupted from peripheral receptive fields, they may play important roles in nerve regeneration and spontaneous pain after nerve injury. These findings are consistent with previous results obtained with snRNA-seq, which showed a new transcriptional state in DRG neurons after spared nerve injury and in trigeminal ganglion neurons after infraorbital nerve transection (*Nguyen et al., 2019*). A similar phenomenon was noted after spinal nerve transection, sciatic nerve transection, and crush injury (*Renthal et al., 2020*; *Wang et al., 2021*). The common characteristics of these new neuronal clusters are the increased expression of injury-induced genes and diminished original

neuron subtype-specific marker genes such as *Tac1*, *Nefh*, *Nppb*, *Mrgprd*, *Th*, and *Sst*, which may represent a general adaptation mechanism after mechanical/traumatic nerve injuries. We focused on examining transcriptional changes at day 7 post-CCI when neuropathic pain-like behavior reaches the peak and enters the maintenance phase. Since gene expression changes vary at different neuropathic pain stages, a time course study of transcriptional changes after CCI is warranted.

*Atf3* induction may be rapid and is frequently detected in isolated cells due to the dissociation process. Unlike *Atf3*, *Sprr1a* showed high specificity as an injury indicator in our scRNA-seq, which is in line with the previous findings (*Nguyen et al., 2017*). Even though *Sprr1a* may be a better standard than *Atf3* to differentiate injured and uninjured neurons, a small number of *Sprr1a*+neurons and CCI-ind1-4 clusters also appeared in sham groups. Their presence may have been induced by the skin incision and muscle damage of sham surgery, which can produce minor nerve injury, or by the cell stress and injury that occur during tissue harvesting and processing (e.g., dissociation, enzymatic digestion). A previous study showed that even sham surgery and minor peripheral injury may increase *Atf3*, *Sox11*, *Sema6a*, *Csf1*, and *Gal* in DRG neurons (*Nguyen et al., 2019*). Importantly, our RNAscope in situ hybridization study also showed that *Sprr1a* expression was minimal in sham group, but remarkably increased after CCI in the injured DRG neurons as indicated by the loss of cell subtype markers. In contrast, uninjured neurons which express the cell subtype markers examined in current study (*Tac1*, *Nefh*, *Nppb*) rarely colocalize with *Sprr1a*. Moreover, the expression of *Fgf3* which is among the top upregulated genes in CCI-ind1-4 clusters also robustly increased after CCI and highly colocalized with *Sprr1a*. Collectively, these findings further suggest that *Sprr1a* may represent an injury indicator and supports our scRNA-seq results.

Another salient finding is that a portion of injured neurons (*Sprr1a*+) in NP1, PEP5, NF1, and NF2 clusters maintained their original neuronal identities at day 7 after CCI, and hence can still be clustered together with uninjured neurons from the same subtype. In contrast, the other eight clusters contained only uninjured neurons after CCI. The injured neurons in these clusters may have lost their identities and were assigned to CCI-ind1-4 clusters. This notion is supported by decreased cell populations in these clusters after CCI. Due to the loss of marker genes for different clusters, the neurochemical identity of CCI-ind1-4 clusters is difficult to confirm. However, a recent snRNA-seq study employed an elegant method to trace back the originalities of 'injured state' neuronal clusters, by examining multiple post-injury time points to consecutively capture residual transcriptional signatures (i.e., a set of processed transcripts that are expressed in a specific cluster) during the transition from uninjured to injured states (*Renthal et al., 2020*). Their findings also suggested that the CCI-induced clusters may arise from neurons with different neurochemical identities before injury. The correlation analysis showed that our CCI-ind1-4 clusters correlated well with clusters such as cLTMR1, cLTMR2, and NP in Renthal's datasets (7 days after sciatic nerve crush). It remains to be determined why injured neurons in NP1, PEP5, NF1, and NF2 clusters can still be clustered together with uninjured neurons from the same subtype, even though they have also lost some subtype-specific marker genes. This apparent discrepancy may be due to these injured neurons losing expression of only a subset of marker genes after CCI, but maintaining the major transcriptional signatures of the naïve state.

Increasing evidence has suggested that uninjured DRG neurons also play important roles in neuropathic pain and show robust neurochemical and functional changes after nerve injury (*Kalpachidou et al., 2022*; *Obata et al., 2003*; *Pertin et al., 2005*; *Tran and Crawford, 2020*). Evoked pain hypersensitivities are common and important neuropathic pain manifestations and are mediated by uninjured neurons through the remaining peripheral innervations. Thus, identifying transcriptional changes in uninjured neurons in a cell-type-specific manner will be important to search for new targets for neuropathic pain treatment. Sciatic nerves contain axons from multiple lumbar DRGs, and CCI causes partial injury to the sciatic nerve. Accordingly, each of these lumbar DRGs contains a mixture of injured and uninjured neurons in CCI model. Strikingly, our analysis showed for the first time that uninjured (*Sprr1a*-) neurons in NP1, PEP5, NF1, and NF2 clusters also undergo subtype-specific changes in gene expression after CCI, as compared to those in the sham group. GO analysis showed that these clusters shared changes in pathways involved in protein biosynthesis (e.g., translation, ribosomal small subunit assembly) with injured neurons, suggesting that they may enter a 'preparation' state in response to nerve injury. Functional changes of uninjured neurons under neuropathic conditions may extend beyond transcriptional regulation to also involve mechanisms such as increased translational rate (*Gebauer and Hentze, 2004*) and modulations by miRNA. In line with this notion, our GO analysis

showed that a pathway named '*positive regulation of Pri-miRNA transcription from RNA polymerase promoter*' was significantly regulated in the NF2 cluster. Systematic investigations are needed to sort out the functional effects of transcriptomic changes in injured and uninjured neurons from each cluster on nerve regeneration, neuronal excitability, and pain.

Renthal et al. also examined co-mingling, uninjured neurons using a sciatic crush injury model. However, they did not find cell-type-specific changes in these neurons. The reason for this discrepancy may be partially due to differences in the techniques (e.g., tissue processing, cell sorting, sequencing depth) and animal models. Compared to CCI model induced by loose ligation of the sciatic nerve, crush injury would injure more nerve fibers and it was estimated that >50% of lumbar DRG neurons are axotomized in this model (**Renthal et al., 2020**). Therefore, the remaining uninjured neurons for sequencing may be much less than that in the CCI model. In addition, we used *Pirt^EGFPf* mice to establish a highly efficient purification approach to enrich neurons for scRNA-seq and therefore largely increased the number of genes detected in DRG neurons. Comparatively, the neuronal selectivity and number of genes detected were lower in the previous studies, which may have resulted in decreased ability to detect these changes and fewer DEGs.

Previous studies suggested that increased *Fgf3* expression in DRG neurons may correlate with pain hypersensitivity (**Brifault et al., 2020**). Here, by examining the functional role of *Fgf3* as a proof-of principle, our calcium imaging study showed that attenuating the increased *Fgf3* expression in DRG neurons of CCI mice with siRNA rather enhanced neuronal responses to capsaicin. This finding suggests that upregulation of *Fgf3* expression after CCI may be an adaptive change that counteracts the development of neuronal hyperexcitability. *Fgf3* was shown to drive neurogenesis of Islet1-expressing motor neurons and mediate axonogenesis in cMet-expressing motor neurons after spinal cord injury in zebrafish (**Goldshmit et al., 2018**). Additional gain-of function and loss-of function studies are needed to further ascertain roles of *Fgf3* in DRG neuron excitability, neuropathic pain and peripheral nerve regeneration, and to explore the underlying mechanisms such as regulating TRPV1 expression or function. It is important for future studies to examine the functional roles of other top DEGs in CCI-ind1-4 clusters and in uninjured neurons, in order to identify new targets for cell-type-specific treatment to promote nerve regeneration and inhibit neuropathic pain.

There is a growing body of literatures on sex differences in neuropathic pain mechanisms (**Machelska and Celik, 2016**). *Xist* is known to be expressed exclusively by the inactive X chromosome in mammals (**Borsani et al., 1991**). Indeed, we also found that it was selectively expressed in female but not male datasets, suggesting that *Xist* is a reliable female marker gene for scRNA-seq. Pearson correlation analysis suggested a similarity of gene expression programs in the two sexes under physiologic conditions. However, we found sex differences in transcriptional changes after CCI, and the cLTMR cluster may play an important role in the sexually dimorphic pain response. Among 296 female and 303 male DEGs induced by CCI, 79 were female-specific, and 86 were male-specific. In addition, 106 genes showed more than twofold differences between the two sexes after CCI. These findings are consistent with a bulk RNA-seq study of DRG, which showed vast differences in genes enriched in pain-relevant pathways between female and male rats after CCI (**Stephens et al., 2019**). Although these findings suggest that peripheral neuronal mechanisms may also underlie sexual dimorphisms in neuropathic pain, **Renthal et al., 2020** reported no differences in injury-induced transcriptional changes between males and females after sciatic nerve crush injury . The reasons for this discrepancy are unclear but may also be due to aforementioned differences in the techniques and animal models, such as the lower read count and numbers of genes detected which may have resulted in decreased ability to detect subtle changes and fewer DEGs in the previous study. Details of the distinct gene networks that function in different neuronal subtypes and might underpin sexual dimorphisms in neuropathic pain must still be explored at a single-cell level in the future.

## Conclusions

In summary, our findings in a well-established animal model of neuropathic pain share some similarities with recent findings in transection injury models, including the loss of marker genes in injured neurons and the emergence of new, injury-induced clusters. Importantly, we demonstrated that subtype-specific transcriptomic changes occurred in both injured and uninjured neurons of NP1, PEP5, NF1, and NF2 clusters after CCI. Furthermore, we also provided novel evidence at single-cell level that transcriptomic sexual dimorphism may occur in DRG neurons after nerve injury, and cLTMR

may play a pivotal role in sex-specific pain modulation. Lastly, by examining *Fgf3* as proof-of-principle, our RNAscope in situ hybridization study provided further evidence supporting findings from scRNA-seq analysis, and in vitro calcium imaging study offered novel insights of the functional implication of *Fgf3*. New knowledge gained from current work and other recent scRNA-seq studies may provide important rationales for developing cell-subtype-specific and sex-based therapies that can optimize neuropathic pain inhibition and sensory nerve regeneration. For instance, an ideal therapy would target top DEGs in uninjured neurons in NP1, PEP5, NF1, and NF2 clusters for pain control, but would not interfere with important transcription programs needed for the regeneration of injured neurons.

## Materials and methods

### *Pirt*<sup>EGFPf</sup> mouse

The *Pirt*<sup>EGFPf</sup> mouse strain (C57BL/6 background) was a gift from Dr. Xinzhong Dong in the Solomon H Snyder Department of Neuroscience, School of Medicine, Johns Hopkins University. The axonal tracer farnesylated enhanced green fluorescent protein (EGFPf) replaced the entire open reading frame of *Pirt* and the EGFPf expression is under the control of the endogenous *Pirt* promoter (*Kim et al., 2008*). We bred the homozygous mutant with WT mice to get heterozygous for experiments. Adult mice (7–8 weeks of age) of both sexes were used. Genotypes of the mice were determined by PCR. Mice were housed three to five per cage and given access ad libitum to food and water. All animal experiments were conducted in accordance with the protocol approved by the Institutional Animal Care and Use Committee of Johns Hopkins University.

### Bilateral sciatic nerve CCI

Contralateral changes may develop after unilateral CCI of the sciatic nerve, including spontaneous pain and mechanical hypersensitivity in hind paws (*Paulson et al., 2002*; *Wilkerson et al., 2020*), as well as gene expression in the spinal cord and DRGs (*Jancálek et al., 2010*). Because transcriptional changes in contralateral DRGs may differ from those on the ipsilateral side, we performed bilateral sciatic CCI to allow the pooling of bilateral DRG tissues for sequencing and to avoid sample variations. The bilateral CCI model has been validated in previous studies, which showed that animals displayed prolonged cold and mechanical hypersensitivities in hind paws but did not exhibit the asymmetric postural or motor influences of unilateral CCI or behavior changes (*Dai et al., 2014*; *Datta et al., 2010*; *Vierck et al., 2005*).

Adult *Pirt*<sup>EGFPf</sup> mice were randomly assigned to undergo bilateral CCI surgery or sham surgery. All procedures were performed by the same experimenter to avoid variation in technique. CCI of the sciatic nerve was performed as previously described (*Guan et al., 2010*; *Li et al., 2017*). Briefly, mice were anesthetized with 2% isoflurane, and a small incision was made at the mid-thigh level. The sciatic nerve was exposed by blunt dissection through the biceps femoris. The nerve trunk proximal to the distal branching point was loosely ligated with three nylon sutures (9–0 nonabsorbable monofilament; S&T AG) placed approximately 0.5 mm apart until the epineuria was slightly compressed and hindlimb muscles showed minor twitching. The muscle layer was closed with a 4–0 silk suture and the wound was closed with metal clips. *Pirt*<sup>EGFPf</sup> mice developed mechanical hypersensitivity in both hind paws, as indicated by a significant increase in paw withdrawal frequency (PWF) to von Frey filament stimulation (*Figure 1B*).

### Mechanical hypersensitivity test

Animals were allowed to acclimate for a minimum of 48 hr before any experimental procedures. Hypersensitivity to punctuate mechanical stimuli was assessed by the PWF method using two von Frey monofilaments (low-force, 0.07 g; high-force, 0.4 g). Each von Frey filament was applied perpendicularly to the mid-plantar area of each hind paw for ~1 s. The left hind paw was stimulated first, followed by the right side (>5 min interval). The stimulation was repeated 10 times at a rate of 0.5–1 Hz (1–2 s intervals). If the animal showed a withdrawal response, the next stimulus was applied after the animal resettled. PWF was then calculated as (number of paw withdrawals/10 trials)×100%.

## Single-cell dissociation

Bilateral L4-5 DRGs were collected from mice at day 7 after bilateral sciatic CCI or sham surgery. Male and female mice from the same litter were subjected to the same surgery (sham or CCI). Bilateral L4-5 DRGs were collected from each mouse and DRGs from five mice of the same group (20 DRGs in total) were pooled as one sample for sequencing. The four groups included Female-Sham, Male-Sham, Female-CCI, and Male-CCI. DRGs were dissected out, digested with 1 mg/mL type I collagenase (Thermo Fisher Scientific) and 5 mg/mL dispase II (Thermo Fisher Scientific) at 37°C for 70 min (10 DRGs/tube), and disassociated into single cells in Neurobasal medium containing 1% bovine serum albumin (BSA). Cells were filtered through a 40 µm cell strainer and centrifuged at 500×$g$ for 5 min. The pellet was resuspended in 500 µL of Triple Express (Thermo Fisher Scientific) and digested at 37°C for 2 min. The reaction was stopped with 1 mL of fetal bovine serum. The cell suspension was laid onto 5 mL of Neurobasal medium containing 20% Percoll and centrifuged at 1400 RPM for 8 min. The upper layer was removed first and then the lower layer. The pellet was washed with 2 mL of Neurobasal medium containing 1% BSA and centrifuged at 500×$g$ for 5 min. The pellet was resuspended with 500 µL of Neurobasal medium containing 1% BSA. The GFP$^+$ cells were sorted into 500 µL of Neurobasal medium containing 1% BSA and centrifuged at 500×$g$ for 5 min to remove most supernatant. The pellet was resuspended with the remaining supernatant to a concentration of ~1000 cells/µL.

## 10× Genomics library preparation and sequencing

The single-cell suspensions were further processed with Chromium Next GEM Single Cell 3' GEM, Library & Gel Bead Kit v3 (PN-1000094) according to the manufacturer's instructions to construct the scRNA-seq library. All libraries were sequenced with the Illumina NovaSeq platform. The raw sequencing reads were processed by Cell Ranger (v.2.1.0) with the default parameters. The reference genome was mm10.

## Single-cell RNA-seq data analysis

Scrublet with default parameters was used first to remove single-cell doublets. After doublet removal, we filtered out cells with fewer than 1500 genes expressed and cells with more than 10% mitochondrial UMI counts. The rigorous filtering was intended to remove smaller residual non-neuronal cells such as SGCs. After doublet removal and quality control, we applied Seurat's integration workflow to correct possible batch effects for the remaining cells of the four datasets. Before the integration, the four datasets were transformed into four individual Seurat objects with standard steps including 'CreateSeuratObject', 'NormalizeData', and 'FindVariableFeatures'. Subsequently, we used 'FindIntegrationAnchors' with the top 3000 variable genes to locate possible anchors among the four datasets. Next, 'IntegrateData' was used to merge the four individual datasets. After the integration, default clustering steps embedded in Seurat were performed with 20 principal components (PCA) and 3000 variable genes. The steps included scaling normalized UMI counts with 'ScaleData', dimensional reduction with 'RunPCA', building a k-nearest neighbor graph with 'FindNeighbors', and finding clusters with the Louvain algorithm by 'FindClusters'. Finally, we visualized identified clusters with 2D UMAP by 'RunUMAP'. To find the most conserved markers in every cluster, we used 'FindConservedMarkers' and show the top 50 markers in *Figure 1F* and *Supplementary file 1*. To evaluate similarities between identified single-cell clusters, we applied unsupervised hierarchical clustering with the pairwise Pearson correlation using 3000 variable genes (*Figure 1—figure supplement 1B*). To find DEGs between clusters or conditions, we used 'FindMarkers' with padj <0.05 and Log$_2$fold-change >0.5 as the thresholds.

## Classification of *Sprr1a*$^+$ and *Sprr1a*$^-$ neurons

We used the expression level of *Sprr1a* to divide neuronal clusters into two subpopulations (*Sprr1a*$^+$and *Sprr1a*$^-$). *Sprr1a*$^+$represent cells with normalized UMI >1 and *Sprr1a*$^-$ represent cells with normalized UMI <1.

## GO analysis

GO analysis was conducted with DAVID (*Huang et al., 2009a*; *Huang et al., 2009b*). We used p-value = 0.05 as the threshold to find enriched GO terms such as biological processes. The gene clusters were visualized with the ClusterProfiler package in R.

## PPI analysis

A PPI network was drawn with the igraph R package based on the list of reported 1002 PPIs involved in pain (*Jamieson et al., 2014*). The genes in the DEG lists that had no connections (receive or send) in the PPI network were filtered out. Removed nodes also filtered out any of their edges.

## RNAscope in situ hybridization

RNAscope fluorescence in situ hybridization experiment was performed according to the manufacturer's instructions, using the RNAscope Multiplex Fluorescent Reagent Kit v2 (ACD, Advanced Cell Diagnostics, Newark, CA) for fresh frozen tissue. Briefly, lumbar DRGs (L4-5) were dissected at 7 days after sham surgery or CCI, frozen, and sectioned into 12 μm sections using a cryostat. In situ probes against the following mouse genes were ordered from ACD and multiplexed in the same permutations across quantified sections: *Sprr1a* (Cat No. 426871-C2), *Fgf3* (Cat No. 503101), *Tac1* (Cat No. 410351-C3), *Nefh* (Cat No. 443671-C3), *Nppb* (Cat No. 425021-C3). High-resolution images of 10 Z-stack were obtained using a ×40 oil immersion objective on Zeiss LSM 800 confocal microscope (Zeiss, Oberkochen, Germany). Cells were considered positive if at least three puncta were observed (*Peirs et al., 2021*).

## Nucleofection

To transfect RNA oligos into DRG neurons, the dissociated neurons from lumbar DRGs were centrifuged to remove the supernatant and resuspended in 100 μL of Amaxa electroporation buffer for mouse neuron (Lonza Cologne GmbH, Cologne, Germany) with siRNAs (0.2 nmol per transfection). si*Fgf3* (sense: CAGAGACCUUGGUACGUGUtt; Antisense: ACACGUACCAAGGUCUCUGgg). Suspended cells were then transferred to a 2.0 mm cuvette and nucleofected with the Amaxa Nucleofector apparatus. After electroporation, cells were immediately mixed to the desired volume of prewarmed culture medium and plated on precoated coverslips or culture dishes. After neurons fully attached to the coverslips or culture dishes (4–6 hr), the culture medium was changed to remove the remnant electroporation buffer.

## Quantitative PCR

To analyze the mRNA expression in DRG neurons, total RNA was isolated using PicoPure RNA Isolation Kit (Thermo Fisher Scientific) following the manufacturer's manual. RNA quality was verified using the Agilent Fragment Analyzer (Agilent Technologies, Santa Clara, CA). Two-hundred ng of total RNA was used to generate the cDNA using the SuperScript VILO MasterMix (Invitrogen, Waltham, MA). Ten ng of cDNA was run in a 20 μl reaction volume (triplicate) using PowerUp SYBR Green Master Mix to measure real-time SYBR green fluorescence with QuantStudio 3 Real-Time PCR Systems (Thermo Fisher Scientific). Calibrations and normalizations were performed using the $2^{-\Delta\Delta CT}$ method. Mouse *Gapdh* was used as the reference gene. Mouse *Fgf3* (#MP204816) and *Gapdh* (#MP205604) primers were purchased from OriGene Technologies (Rockville, MD). *Fgf3*: (forward primer: GCAAGCTCTACT GCGCTACCAA; reverse primer: CACTTCCACCGCAGTAATCTCC). *Gapdh* (forward primer: CATC ACTGCCACCCAGAAGACTG; reverse primer: ATGCCAGTGAGCTTCCCGTTCAG).

## DRG neuronal culture and in vitro calcium imaging

Experiments were conducted as that in our previous studies (*Liu et al., 2009*). Lumbar DRGs from 4-week-old mice that underwent sham surgery or CCI were collected in cold DH10 20 (90% DMEM/F--12, 10% fetal bovine serum, penicillin [100 U/mL], and streptomycin [100 μg/mL] Invitrogen, Waltham, MA) and treated with enzyme solution (dispase [5 mg/mL] and collagenase type I [1 mg/mL] in Hanks' balanced salt solution without $Ca^{2+}$ or $Mg^{2+}$) for 35 min at 37°C. After trituration, the supernatant with cells was filtered through a Falcon 40 μm (or 70 μm) cell strainer. Then, the cells were spun down with centrifugation and were resuspended in DH10 with growth factors (25 ng/mL NGF; 50 ng/mL GDNF), plated on glass coverslips coated with poly-D-lysine (0.5 mg/mL; Biomedical Technologies Inc, Madrid,

Spain) and laminin (10 µg/mL; Invitrogen), cultured in an incubator (95% $O_2$ and 5% $CO_2$) at 37°C, and used within 48 hr. Neurons were loaded with Fura-2-acetomethoxyl ester (Molecular Probes, Eugene, OR) for 45 min in the dark at room temperature (*Liu et al., 2009*). After being washed, cells were imaged at 340 and 380 nm excitation for the detection of intracellular free calcium.

### Integrated analysis of our scRNA-seq datasets and Renthal's snRNA-seq datasets of mouse DRG neurons

We conducted the integration analysis of our scRNA-seq datasets and Renthal's snRNA-seq datasets (GSE154659). We extracted DRG cells from naïve mice (seven replicates: GSM4676529, GSM4676530, GSM4676533, GSM4676534, GSM4676535, GSM4676536, GSM4676537) and mice at day 7 after sciatic nerve crush injury (four replicates: GSM4676564, GSM4676565, GSM4676531, GSM4676532) from Renthal's datasets. We only considered the cells with gene count between 200 and 12,000, and mitochondrial DNA less than 10%. The remaining 17,207 cells were integrated with our datasets using default parameters described in Seurat's integrated workflow. The first 30 PCA were used to build UMAP for visualizing the integration result.

### Correlation analysis between our identified neuronal clusters and neuronal clusters of Renthal's datasets

We extract cells (four replicates: GSM4676564, GSM4676565, GSM4676531, GSM4676532) from mice at day 7 after sciatic nerve crush injury from Renthal's datasets and then used Seurat's default workflow to create its Seurat single-cell object. We further used shared variable genes between our dataset and their datasets to calculate Pearson correlation.

### Data and statistical analysis

Statistical analyses were performed with the Prism 8.0 statistical program (GraphPad Software, Inc). The methods for statistical comparisons in each study are given in the figure legends. Comparisons of data consisting of two groups were made by Student's t-test. Comparisons of data in three or more groups were made by one-way analysis of variance (ANOVA) followed by the Bonferroni post hoc test. Comparisons of two or more factors across multiple groups were made by two-way ANOVA followed by the Bonferroni post hoc test. Two-tailed tests were performed, and $p < 0.05$ was considered statistically significant in all tests.

## Acknowledgements

This study was conducted at the Johns Hopkins University School of Medicine. The authors thank Claire F Levine, MS (scientific editor, Department of Anesthesiology and Critical Care Medicine, Johns Hopkins University), for editing the manuscript. This study was supported by the National Institutes of Health (Bethesda, MD) grants NS070814 (YG), NS110598 (YG), NS117761 (YG), AG066603 (XC), and AG0689997 (XC, YG), and was facilitated by the Pain Research Core, which is funded by the Blaustein Fund and the Neurosurgery Pain Research Institute at the Johns Hopkins School of Medicine. Xue-Wei Wang was supported by an NIH grant K99EY031742. The authors thank Dr. Xinzhong Dong for sharing the *Pirt*[EGFPf] mouse strain. The authors thank Hao Zhang from the Flow Cytometry Cell Sorting Core Facility at Bloomberg School of Public Health, Johns Hopkins University for doing FACS sorting. The facility was supported by 1S10OD016315-01,1S10RR13777001, and in part by CFAR: 5P30AI094189-04 (Chaisson). The authors appreciate the Johns Hopkins Single Cell and Transcriptomics Core for conducting the scRNA-seq. Funders had no role in study design, data collection, data interpretation, or in the decision to submit the work for publication. The authors declare no competing interests. There are no other relationships that might lead to a conflict of interest in the current study.

# Additional information

### Funding

| Funder | Grant reference number | Author |
|---|---|---|
| National Institutes of Health | NS070814 | Yun Guan |
| National Institutes of Health | NS110598 | Yun Guan |
| National Institutes of Health | NS117761 | Yun Guan |
| National Institutes of Health | AG0689997 | Xu Cao<br>Yun Guan |
| National Institutes of Health | K99EY031742 | Xue-Wei Wang |
| National Institutes of Health | AG066603 | Xu Cao |
| Johns Hopkins University | | Yun Guan |

The funders had no role in study design, data collection and interpretation, or the decision to submit the work for publication.

### Author contributions

Chi Zhang, Conceptualization, Data curation, Formal analysis, Investigation, Visualization, Methodology, Writing - original draft, Writing - review and editing; Ming-Wen Hu, Data curation, Formal analysis, Investigation, Visualization, Methodology, Writing - original draft, Writing - review and editing; Xue-Wei Wang, Data curation, Methodology, Writing - review and editing; Xiang Cui, Validation, Visualization; Jing Liu, Methodology; Qian Huang, Project administration; Xu Cao, Conceptualization, Investigation, Writing - review and editing; Feng-Quan Zhou, Conceptualization, Supervision, Writing - review and editing; Jiang Qian, Supervision, Writing - review and editing; Shao-Qiu He, Formal analysis, Visualization, Writing - review and editing; Yun Guan, Conceptualization, Resources, Supervision, Funding acquisition, Validation, Investigation, Writing - original draft, Project administration, Writing - review and editing

### Author ORCIDs

Chi Zhang http://orcid.org/0000-0001-7306-2243
Xue-Wei Wang http://orcid.org/0000-0002-1375-7358
Xu Cao http://orcid.org/0000-0001-8614-6059
Shao-Qiu He http://orcid.org/0000-0001-9490-6986
Yun Guan http://orcid.org/0000-0003-1321-6655

### Ethics

This study was performed in strict accordance with the recommendations in the Guide for the Care and Use of Laboratory Animals of the National Institutes of Health. All animal experiments were conducted in accordance with the protocol approved by the Institutional Animal Care and Use Committee (IACUC, protocol #: MO19M308, PI: Yun Guan) of the Johns Hopkins University. All surgery was performed under sufficient inhalation anesthesia, and every effort was made to minimize suffering.

### Decision letter and Author response

Decision letter https://doi.org/10.7554/eLife.76063.sa1
Author response https://doi.org/10.7554/eLife.76063.sa2

# Additional files

### Supplementary files

- Supplementary file 1. Top 50 conserved marker genes in 16 neuronal clusters (related to *Figure 1*).
- Supplementary file 2. Quality control metrics (related to *Figure 1*).

• Supplementary file 3. Differentially expressed genes in NP1, PEP5, NF1, and NF2, comparing chronic constriction injury (CCI) with Sham (padj <0.05) (related to *Figure 4*).

• Supplementary file 4. Nineteen differentially expressed genes shared by NP1, PEP5, NF1, and NF2, comparing chronic constriction injury (CCI) with Sham (related to *Figure 4*).

• Supplementary file 5. Differentially expressed genes in Sprr1a⁻ neurons of NP1, PEP5, NF1, and NF2, comparing chronic constriction injury (CCI) with Sham (padj <0.05) (related to *Figure 5*).

• Supplementary file 6. Enriched pathways after chronic constriction injury (CCI) in NP1, PEP5, NF1, and NF2 with the corresponding genes (related to *Figure 5*).

• Supplementary file 7. Lists of chronic constriction injury (CCI)-induced differentially expressed genes (DEGs) including DEGs only in males, DEGs only in females, and DEGs shared by both males and females (padj <0.05, $Log_2$fold-change >0.5) (related to *Figure 6*).

• Supplementary file 8. Enriched pathways with the 106 chronic constriction injury (CCI)-induced differentially expressed genes (DEGs) (related to *Figure 6*).

• Transparent reporting form

## Data availability

Sequencing data is deposited in GEO under accession number GSE216039.

The following dataset was generated:

| Author(s) | Year | Dataset title | Dataset URL | Database and Identifier |
|---|---|---|---|---|
| Zhang C, Hu M-W, Wang X-W, Cui X, Liu J, Huang Q, Cao X, Zhou F-Q, Qian J, S-Q He, Guan Y | 2022 | scRNA-sequencing reveals subtype-specific transcriptomic perturbations in DRG neurons of PirtEGFPf mice in neuropathic pain condition | http://www.ncbi.nlm.nih.gov/geo/query/acc.cgi?acc=GSE216039 | NCBI Gene Expression Omnibus, GSE216039 |

The following previously published dataset was used:

| Author(s) | Year | Dataset title | Dataset URL | Database and Identifier |
|---|---|---|---|---|
| Renthal W, Tochitsky I, Yang L, Cheng YC, Li E, Kawaguchi R, Geschwind DH, Woolf CJ | 2020 | Transcriptional Reprogramming of Distinct Peripheral Sensory Neuron Subtypes after Axonal Injury | https://www.ncbi.nlm.nih.gov/geo/query/acc.cgi?acc=GSE154659 | NCBI Gene Expression Omnibus, GSE154659 |

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
