## [Editor Report]

This study identifies injury-induced changes in transcriptomic signatures in peripheral sensory neurons at a single cell level, revealing key insights into sexual dimorphism as well as plasticity-related differences between injured and uninjured neurons. These results promote an understanding of the nature of molecular events involved in the establishment of neuropathic pain.

---

## [Decision Letter]

**Decision letter after peer review:**

Thank you for submitting your article "scRNA-sequencing reveals subtype-specific transcriptomic perturbations in DRG neurons of *Pirt-EGFPf* mice in neuropathic pain condition" for consideration by *eLife*. Your article has been reviewed by 2 peer reviewers, one of whom is a member of our Board of Reviewing Editors, and the evaluation has been overseen by Mone Zaidi as the Senior Editor. The following individual involved in review of your submission has agreed to reveal their identity: Theodore J. Price (Reviewer #2).

Essential revisions:

Four issues were brought up in the review.

1. It was felt that there is limited novelty as single cell RNA seq datasets had been published previously. Please discuss you findings in relation to other papers.

2. The newly-identified gene candidates have not been validated for functionality.

Knockdown experiments will strengthen this aspect.

3. As noted below, there is the issue of separating injured from uninjured cells, which, once again, needs to be worked on or discussed in depth.

4. All comments under "Recommendations to Authors" need to be addressed.

The request for a later time point is not considered as an essential revision.

*Reviewer #1 (Recommendations for the authors):*

Despite solid insights, the level of mechanistic advance and in-depth understanding represented by the results in the current form is limited. For example, although it is a major benefit to separate injured and uninjured neurons, axonal regeneration and plasticity-related genes were reported in both types. This is similar to previous bulk sequencing data which reports regulation of a vast number of injury-related genes.

Moreover, sexual dimorphism was noted here in contrast to a previous study on single cell RNAseq, but this has already been reported in previous bulk sequencing data from the same model of neuropathic pain.

Most importantly, the data remain at a descriptive level and exclusively involve data analyses on transcriptomic experiments. There are no follow-up analyses on the functional significance of the findings. Given that at least 3 single cell transcriptomics studies have been already published on the DRG after nerve injury, it would be of crucial importance to take up some of the most interesting transcriptional changes, such as in the new clusters described here or in injured vs. uninjured neurons, and study their functional impact on excitability of peripheral sensory neurons and neuropathic pain.

An additional experiment which would enhance the value of the study in revealing core mechanisms of neuropathic pain and particularly help delineate which transcriptomic signatures pertain to injury-related axonal structural changes vs. the emergence of neuropathic pain would be to perform single cell RNAseq after hyperexcitability is reduced and normal pain sensitivity is established. This is possible with the CCI model, since, as opposed to the spared nerve injury model employed in other transcriptomics studies, hypersensitivity and neuropathic pain are reversed within a few weeks post-injury.

Other Points:

Introduction: Sprr1a is indicated to be selectively expressed in injured neurons and employed as a marker for distinguishing these from uninjured neurons, but no citation provided.

In methods, please state how many DRGs from how many different mice were pooled to constitute single single samples for sequencing.

In order for the readers to judge the quality of the sequencing data, please show in methods or supplementary information how many cells showed less than 1500 genes and what proportion of the raw sequencing data was represented by doublets.

At the beginning of the Results section on sequencing data, please indicate the final n number of mice used for sequencing and the n number of samples. For behavioural experiments, an n of 5 mice/sex is indicated. Since DRGs were pooled across mice, does this mean that the n number for sequencing was less than 5?

Figure 1: The color coding in Figure 1C does not allow to clearly delineate the 4 different experimental conditions shown. The shades CCI and sham are hardly different and cannot be distinguished at the size shown. Please change to distinct colors. The data also do not enable determining whether the putative new CCI clusters are only seen in CCI mice. In Figure 2c, for example, one can see a similar cluster, albeit less dense, in sham-treated mice.

What is the neurochemical identity of the new CCI clusters? The Results section states that these neurons likely emerge from other cellular clusters owing to a loss of expression of typical markers. Since this is an important finding of this manuscript, the study should study the neurochemical properties and origins of these neurons. At least some of the genes induced in transcriptomics analyses should be validated at protein level and/or at least at mRNA level using in situ hybrisidation.

In the discussion, it is stated that Pirt1 is expressed in 'most' DRG neurons. Thus, are there DRG neurons that do not express Pirt1? It is important to transparently present this information since it is highly pertinent to the interpretation of the data.

*Reviewer #2 (Recommendations for the authors):*

I have a comment with respect to adding new experiments, in addition to the public critique. I think that the paper is relatively weak compared to similar published papers on transcriptomic changes in the DRG after nerve injury for 2 reasons:

1) The authors focused on only a single time point. The paper would be far better if the transcriptomes of DRG neurons were evaluated at a late time point, and after pain had resolved. As mentioned above, this is the strength of the model used here.

2) The authors have not really used their data to gain new insight. For instance, they suggest that the NF1 and NF2 neurons might be involved in key aspects of neuropathic pain, but they have not used the information gained from the transcriptomes of these cells to do any pharmacological or genetic experiments to expand the insight.

Addressing one of these issues with new experiments would greatly improve the paper, at least in my opinion.

[Editors’ note: further revisions were suggested prior to acceptance, as described below.]

Thank you for resubmitting your work entitled "scRNA-sequencing reveals subtype-specific transcriptomic perturbations in DRG neurons of *Pirt^EGFPf^* mice in neuropathic pain condition" for further consideration by *eLife*. Your revised article has been evaluated by Mone Zaidi (Senior Editor) and a Reviewing Editor.

The manuscript has been improved but there are some remaining issues that need to be addressed, as outlined below:

In particular, information on statistical analyses should be included more prominently and clearly in the manuscript. A section should be added under methods, and the figure legends should clearly describe the statistical tests employed. At present, asterisks are included with p values in some figures, but information on the tests employed is missing, while in other figures, statistical information is missing altogether.

---

## [Author Response]

Essential revisions:Four issues were brought up in the review.1. It was felt that there is limited novelty as single cell RNA seq datasets had been published previously. Please discuss you findings in relation to other papers.

Although a few scRNA-seq studies of DRG after injury were published recently, there remain many important questions not fully addressed in previous studies, especially changes related to neuropathic pain. For example, injured and uninjured DRG neurons may play different roles in neuropathic pain. Specifically, since the injured (axotomized) neurons have lost peripheral innervations, they can no longer mediate evoked pain hypersensitivity. Instead, these neurons may develop spontaneous activity, and play an important role in ongoing pain and central sensitization. On the other hand, uninjured neurons still maintain peripheral innervations and mediate evoked pain hypersensitivity. Clinically, both ongoing pain and evoked pain hypersensitivities are important neuropathic pain manifestations, and may require different treatments due to different underlying mechanisms which remain partially known. Thus, identifying and differentiating transcriptional changes in both injured and uninjured neurons in a cell-type-specific manner will be important to search for new targets for treatment. However, to our knowledge, previous scRNA-seq studies have mainly focused on changes in injured neurons. So far, details of subtype-specific transcriptomic changes in uninjured DRG neurons under neuropathic pain conditions remain unclear.

Technically, we developed a new methodology by using Pirt^EGFPf^ mice that selectively express an enhanced green fluorescent protein in DRG neurons, and established a highly efficient purification approach to enrich neurons for single-cell RNA-seq. This approach significantly increased the number of genes detected in DRG neurons, thus unraveling more details that have not been shown in previous studies. Our findings extended previous findings of changes in injured neurons. Importantly, we unraveled that uninjured DRG neurons of four unique clusters/subtypes exhibited significant transcriptomic perturbations after CCI. Conceptually, these findings of subtype-specific transcriptomic changes in uninjured DRG neurons have not been reported before.

We further applied pain-focused analysis of these data sets and demonstrated neuropathic pain-specific PPI networks of CCI-induced DEGs in NP1, PEP5, NF1, and NF2 clusters (Figure 4B-E). To our knowledge, this has not been done before. In addition, our study provided novel evidence of sex differences in transcriptional changes of different DRG neuronal subtypes after CCI, which have not been shown previously at the single-cell level. Collectively, these findings demonstrate the conceptual novelty of our study.

Finally, in this revision, we added new RNAscope in situ hybridization studies to validate changes in *Fgf3* gene expressions and distribution in DRG neurons. Moreover, we performed functional studies using in vitro calcium imaging and demonstrated the functional impact of *Fgf3* as proof of principle. Thus, we further improved the scientific rigor, innovation, and significance of our study. New knowledge gained from current work and other recent scRNA-seq studies may provide important rationales for developing cell subtype-specific and sex-based therapies that can optimize neuropathic pain inhibition and sensory nerve regeneration.

This research field is developing rapidly, we highlighted the novelty and significance of our findings in the discussion (Page 20), and also discussed about several major publications in the field as suggested by the reviewer. (e.g., Page 21, 22, 23, 25).

2. The newly-identified gene candidates have not been validated for functionality.Knockdown experiments will strengthen this aspect.

We thank the reviewer for this good comment. We agree that functional validation of newly-identified gene candidates is important to extend current findings. The expression of *Fgf3* is among the top upregulated genes in CCI-ind1-4 clusters. Our new in situ hybridization study provided molecular evidence supporting the findings from the scRNA-seq analysis. Accordingly, we conducted an in vitro calcium imaging study to examine the function of upregulated *Fgf3* expression as a proof of principle. Future studies need to examine the functional roles of other top DEGs in CCI-ind1-4 clusters and those in uninjured neurons, to identify new targets for cell-type specific treatment to promote nerve regeneration and inhibit neuropathic pain. We included these new findings in Results (Figure 7, Figure S5, Page 19-20) and also provided a discussion (Pages20, 24-25)

3. As noted below, there is the issue of separating injured from uninjured cells, which, once again, needs to be worked on or discussed in depth.

Sciatic nerves contain axons from multiple lumbar DRGs, and CCI causes partial injury to the sciatic nerve. Accordingly, each lumbar DRG contains a mixture of injured and uninjured neurons. Unlike CCI, the L5 spinal nerve ligation (SNL) would injure most of the neurons in L5 DRG, but neurons in neighboring DRGs were largely uninjured. Although injured and uninjured neurons can be readily segregated in the SNL model, clinical neuropathic pain conditions often resulted from a peripheral nerve injury involving a mixture of injured and uninjured neurons in the same DRG. Importantly, since glia-neuron and neuron-neuron interaction may occur within the same DRG, the excitability and transcriptional changes in uninjured neurons can be greatly affected by neighboring injured neurons. We provided more information in the introduction (Page 5), and discussion of these issues (Pages 22, 23).

4. All comments under "Recommendations to Authors" need to be addressed.

We have fully addressed all comments.

The request for a later time point is not considered as an essential revision.

Thanks for your understanding. We also provide a brief discussion of this limitation on Page 22.

Reviewer #1 (Recommendations for the authors):Despite solid insights, the level of mechanistic advance and in-depth understanding represented by the results in the current form is limited. For example, although it is a major benefit to separate injured and uninjured neurons, axonal regeneration and plasticity-related genes were reported in both types. This is similar to previous bulk sequencing data which reports regulation of a vast number of injury-related genes.Moreover, sexual dimorphism was noted here in contrast to a previous study on single cell RNAseq, but this has already been reported in previous bulk sequencing data from the same model of neuropathic pain.

We appreciate the reviewer’s thoughtful comments and support! We provided a further discussion of these issues (please see details below).

DRG contains functional different and molecular heterogeneous subgroups of sensory neurons and may show different transcriptional and function changes after injury. Sciatic nerves contain axons from multiple lumbar DRGs, and CCI causes partial injury to the sciatic nerve. Accordingly, each lumbar DRG contains a mixture of injured and uninjured neurons. Unlike CCI, the L5 spinal nerve ligation (SNL) would injure most neurons in L5 DRG, but neurons in neighboring DRGs were largely uninjured. Although injured and uninjured neurons can be readily segregated in the SNL model, clinical neuropathic pain conditions often result from a peripheral nerve injury involving a mixture of injured and uninjured neurons in the same DRG. Importantly, since glia-neuron and neuron-neuron interaction may occur within the same DRG, the excitability and transcriptional changes in uninjured neurons can be greatly affected by neighboring injured neurons. Thus, we used CCI model in this study for studying changes in both injured and uninjured neurons after nerve injury. We provided further information in the introduction (Page 5) and discussion (Page 23)

Although some findings of axonal regeneration and plasticity-related genes were similar to those identified in the bulk RNA-seq study, our scRNA-seq study provided much more details and new findings of the cell-type-specific change. This is partially due to our new methodology of using Pirt^EGFPf^ mice that selectively express an enhanced green fluorescent protein in DRG neurons, and established a highly efficient purification approach to enrich neurons for single-cell RNA-seq. This approach significantly increased the number of genes detected in DRG neurons, thus unraveling more details, especially changes in uninjured DRG neurons, which have not been shown in previous studies. For example, we found that uninjured DRG neurons of four unique clusters/subtypes exhibited significant transcriptomic perturbations after CCI. Since peripheral inputs are mediated by uninjured neurons, they mediate evoked pain hypersensitivity through the remaining peripheral innervations. Clinically, evoked pain hypersensitivities are common and important neuropathic pain manifestations, and may require different treatments from ongoing pain due to different underlying mechanisms. Thus, identifying transcriptional changes in uninjured neurons in a cell-type-specific manner will be important to search for new targets for neuropathic pain treatment. To our knowledge, these findings of subtype-specific transcriptomic changes in uninjured DRG neurons cannot be detected by using bulk RNA-seq and have not been reported before.

Regarding the sex difference, our study also provided novel evidence of sex differences in transcriptional changes of different DRG neuronal subtypes after CCI. These findings are consistent with sexual dimorphism shown in previous bulk sequencing data and further substantiated peripheral neuronal mechanisms for sexual dimorphisms in neuropathic pain conditions. Importantly, compared to bulk RNA-seq, the scRNA-seq study provided more details of cell-type-specific information. For example, Figure 6E shows the correlation between male and female datasets is strong in NP, PEP, NF, and CCI-induced clusters, but is only moderate in the cLTMR cluster.

We are aware that sexual dimorphism was not detected in a previous scRNA-seq study (see Discussion Page 25). Renthal et al. reported no differences in injury-induced transcriptional changes between males and females after sciatic nerve crush injury (Renthal et al., 2020). Although the reasons for this discrepancy are unclear, we speculate that it may be partially due to differences in the techniques (e.g., tissue processing, cell sorting, sequencing depth) and animal models (CCI versus crush injury). For example, we used Pirt^EGFPf^ mice to establish a highly efficient purification approach to enrich DRG neurons for single-cell RNA-seq, and therefore largely increased the number of genes detected in DRG neurons. Comparatively, the neuronal selectivity and number of genes detected could be lower in the previous study, which may have resulted in fewer DEGs and decreased ability to detect subtle changes.

We agree that details of the distinct gene networks that function in different neuronal subtypes underlying sexual dimorphisms still need to be explored in the future, and we included a brief discussion (Page 25). Please also refer to our reply to Editor’s comment 1.

Most importantly, the data remain at a descriptive level and exclusively involve data analyses on transcriptomic experiments. There are no follow-up analyses on the functional significance of the findings. Given that at least 3 single cell transcriptomics studies have been already published on the DRG after nerve injury, it would be of crucial importance to take up some of the most interesting transcriptional changes, such as in the new clusters described here or in injured vs. uninjured neurons, and study their functional impact on excitability of peripheral sensory neurons and neuropathic pain.

We agree with the reviewer that functional validation of newly-identified gene candidates is important to extend current findings. Thus, we conducted a new in vitro calcium imaging study to examine the function of upregulated *Fgf3* expression after CCI as a proof-of-principle. *Fgf3* is among the top upregulated genes in CCI-ind1-4 clusters. Our new in situ hybridization study provided molecular and morphological evidence supporting findings from the scRNA-seq analysis. in vitro calcium imaging study offered insights into the functional implications of *Fgf3*. Fully examining the functionality of other top DEGs in CCI-ind1-4 clusters and those in uninjured neurons of NP1, PEP5, NF1 and NF2, especially in vivo on neuropathic pain and nerve regeneration will require a significant amount of work, and will be conducted in the future. We included these new findings in Results (Figure 7, Figure S5, Page 19-20) and also provided a discussion (Pages20, 24-25)

An additional experiment which would enhance the value of the study in revealing core mechanisms of neuropathic pain and particularly help delineate which transcriptomic signatures pertain to injury-related axonal structural changes vs. the emergence of neuropathic pain would be to perform single cell RNAseq after hyperexcitability is reduced and normal pain sensitivity is established. This is possible with the CCI model, since, as opposed to the spared nerve injury model employed in other transcriptomics studies, hypersensitivity and neuropathic pain are reversed within a few weeks post-injury.

We appreciate this good suggestion. However, conducting a new set of scRNA-seq studies at the later post-injury time point when neuropathic pain has diminished would require a significant amount of work and is beyond the scope of this study. We acknowledged this limitation and provided a brief discussion (Page 22).

Other Points:Introduction: Sprr1a is indicated to be selectively expressed in injured neurons and employed as a marker for distinguishing these from uninjured neurons, but no citation provided.

We provide the reference for Sprr1a (Bonilla et al., 2002).

In methods, please state how many DRGs from how many different mice were pooled to constitute single single samples for sequencing.

We provided more information on the methods. Bilateral L4-5 DRGs were collected from each mouse subjected to bilateral CCI or sham surgery of the sciatic nerve. DRGs from five mice (20 in total) were pooled to constitute one single sample for sequencing (Page 7)

In order for the readers to judge the quality of the sequencing data, please show in methods or supplementary information how many cells showed less than 1500 genes and what proportion of the raw sequencing data was represented by doublets.

We provided the information as requested. Please refer to Figure S1B and Table S2.

At the beginning of the Results section on sequencing data, please indicate the final n number of mice used for sequencing and the n number of samples. For behavioural experiments, an n of 5 mice/sex is indicated. Since DRGs were pooled across mice, does this mean that the n number for sequencing was less than 5?

We provided more information. Five mice were used for each sample and 4 samples were sequenced (Page 7).

Figure 1: The color coding in Figure 1C does not allow to clearly delineate the 4 different experimental conditions shown. The shades CCI and sham are hardly different and cannot be distinguished at the size shown. Please change to distinct colors. The data also do not enable determining whether the putative new CCI clusters are only seen in CCI mice. In Figure 2c, for example, one can see a similar cluster, albeit less dense, in sham-treated mice.

We modified the color coding. We did NOT claim that the new CCI-induced clusters are only seen in CCI mice. Compared to the prominent presence in CCI groups, these CCI-induced clusters were minimal in sham groups. We discussed the presence of these clusters at a low level in sham groups in our original Discussion section (Page 22). As the reviewer suggested, this may have been induced by the skin incision and muscle damage of sham surgery, which can produce minor nerve injury, or by the cell stress and injury that occur during tissue harvesting and processing (e.g., dissociation, enzymatic digestion). A previous study showed that even sham surgery and minor peripheral injury may increase Atf3, Sox11, Sema6a, Csf1, and Gal in a small portion of DRG neurons (Nguyen et al., 2019). This information was included in our original Discussion section (Page 22).

What is the neurochemical identity of the new CCI clusters? The Results section states that these neurons likely emerge from other cellular clusters owing to a loss of expression of typical markers. Since this is an important finding of this manuscript, the study should study the neurochemical properties and origins of these neurons. At least some of the genes induced in transcriptomics analyses should be validated at protein level and/or at least at mRNA level using in situ hybrisidation.

Due to the loss of marker genes for different clusters, the neurochemical identity of the new CCI clusters is difficult to confirm. However, a previous study employed an elegant method to trace back the originalities of “injured state” neuronal clusters (Renthal et al., 2020). Their findings suggested that these newly emerged clusters may arise from neurons with different neurochemical identities before injury. We compared our CCI-ind 1-4 clusters with their datasets (7 days after crush) and found that the CCI-ind1-4 clusters showed good correlation with some clusters such as cLTMR1, cLTMR2, and NP in their datasets (Figure S6). We include this finding and more information in the discussion (Page 23).

As the reviewer suggested, we conducted a new in situ hybridization study to examine genes induced in transcriptomics analyses. Our RNAscope in situ hybridization study showed that Sprr1a expression was minimal in the sham group but remarkably increased after CCI. In contrast, uninjured neurons which express the cell subtype markers examined in the current study (Tac1, Nefh, Nppb) rarely colocalize with Sprr1a. Moreover, the expression of *Fgf3* which is among the top upregulated genes in CCI-ind1-4 clusters, also robustly increased after CCI and highly colocalized with Sprr1a. Collectively, the molecular evidence further suggests that Sprr1a may represent an injury indicator, and supports our scRNA-seq results. We included these new findings in Results (Figure 7) and provided a discussion (Page 23)

In the discussion, it is stated that Pirt1 is expressed in 'most' DRG neurons. Thus, are there DRG neurons that do not express Pirt1? It is important to transparently present this information since it is highly pertinent to the interpretation of the data.

This information was presented in the last paragraph of the introduction that Pirt is expressed in >83.9% of neurons in mouse DRG, but not in other cell types (Kim et al., 2008). Specifically, in the previous study done by our colleague Dr. Xinzhong Dong who developed this transgenic mouse line, expression of the knockin EGFPf was under the control of the endogenous *Pirt* promoter. Anti-GFP antibody staining revealed that GFP is widely expressed in the DRG isolated from these mice, labeling 83.9% of all neurons. Interestingly, *Pirt*-negative neurons are mainly NF200^+^ and have large-diameter cell bodies. We included a brief discussion of a potential limitation that the NF population may be slightly underrepresented in our sample (Page 21).

Reviewer #2 (Recommendations for the authors):I have a comment with respect to adding new experiments, in addition to the public critique. I think that the paper is relatively weak compared to similar published papers on transcriptomic changes in the DRG after nerve injury for 2 reasons:1) The authors focused on only a single time point. The paper would be far better if the transcriptomes of DRG neurons were evaluated at a late time point, and after pain had resolved. As mentioned above, this is the strength of the model used here.

We thanks reviewer for this comment. Although meaningful, it was not our intention to conduct a time course study to fully characterize time-dependent transcriptional changes using scRNA-seq, which is costly and require a great effort for data analysis, etc., and is beyond the scope of the current study. We provided a brief discussion of this (Page 21).

2) The authors have not really used their data to gain new insight. For instance, they suggest that the NF1 and NF2 neurons might be involved in key aspects of neuropathic pain, but they have not used the information gained from the transcriptomes of these cells to do any pharmacological or genetic experiments to expand the insight.Addressing one of these issues with new experiments would greatly improve the paper, at least in my opinion.

We agree with the reviewer that functional validation of newly-identified gene candidates is important to extend current findings. Thus, we conducted a new in vitro calcium imaging study to examine the function of upregulated *Fgf3* expression after CCI as a proof-of-principle The expression of *Fgf3* which is among the top upregulated genes in CCI-ind1-4 clusters, our new in situ hybridization study provided morphological evidence supporting findings from the scRNA-seq analysis. in vitro calcium imaging study offered novel insights into the functional implications of *Fgf3*. Fully examining the functionality of other top DEGs in CCI-ind1-4 clusters and those in uninjured neurons, especially in vivo on neuropathic pain and nerve regeneration will require a significant amount of work and will be conducted in the future. We included these new findings in Results (Figure 7, Figure S5, Page 18-20).and also provided a discussion (Pages20, 24-25)

[Editors’ note: further revisions were suggested prior to acceptance, as described below.]

The manuscript has been improved but there are some remaining issues that need to be addressed, as outlined below:In particular, information on statistical analyses should be included more prominently and clearly in the manuscript. A section should be added under methods, and the figure legends should clearly describe the statistical tests employed. At present, asterisks are included with p values in some figures, but information on the tests employed is missing, while in other figures, statistical information is missing altogether.

We appreciate your further reviewing and thoughtful comments! We have carefully addressed all remaining issues (more details of statistical analyses, statistical tests employed and data presentation, etc.), and revised the manuscript.